# ABBEL: LLM Agents Acting through Belief Bottlenecks Expressed in Language

## Abstract

As the length of sequential decision-making tasks increases, it becomes computationally impractical to keep full interaction histories in context. We introduce a general framework for LLM agents to maintain concise contexts through multi-step interaction: Acting through Belief Bottlenecks Expressed in Language (ABBEL), and methods to further improve ABBEL agents with RL post-training. ABBEL replaces long multi-step interaction history by a *belief state*, i.e., a natural language summary of what has been discovered about task-relevant unknowns. Under ABBEL, at each step the agent first updates a prior belief with the most recent observation from the environment to form a posterior belief, then uses only the posterior to select an action. We systematically evaluate frontier models under ABBEL across six diverse multi-step environments, finding that ABBEL supports generating interpretable beliefs while maintaining near-constant memory use over interaction steps. However, bottleneck approaches are generally prone to error propagation, which we observe causing inferior performance when compared to the full context setting due to errors in belief updating. Therefore, we train LLMs to generate and act on beliefs within the ABBEL framework via reinforcement learning (RL). We experiment with belief grading, to reward higher quality beliefs, as well as belief length penalties to reward more compressed beliefs. Our experiments demonstrate the ability of RL to improve ABBEL's performance beyond the full context setting, while using less memory than contemporaneous approaches.

## 1 Introduction

Recent approaches to automating complex tasks such as software development and scientific research result in AI systems that take hundreds or thousands of steps of interaction with their environment, often exceeding the practical context limits of even frontier models. These limitations necessitate the development of methods that compress interaction histories while preserving the most relevant information for effective decision-making. While work on maintaining minimal sufficient statistics for sequential decision-making stretches back to Åström (1965), LLMs provide a unique opportunity for expressing such information in *language*, a medium that is both flexible and interpretable. The information in the interaction history required to solve a task can generally be described by a posterior belief over the values of task-relevant variables. Compressing an interaction history into such a belief state could, in principle, limit the growth of the context length without harming performance. Furthermore, recent work suggests that LLMs can accurately update natural language descriptions of beliefs given new observations (Arumugam & Griffiths, 2025), and prompting language agents to explicitly generate a belief before acting can enhance their performance (Kim et al., 2025).

In light of this, we propose **ABBEL** (**Acting through Belief Bottlenecks Expressed in Language**), a framework for maintaining compact and interpretable contexts where an agent generates and acts on natural language belief states instead of full interaction histories. Figure 1 illustrates ABBEL in the multi-step word guessing game *Wordle*[1]. ABBEL replaces the full history of guesses and feedback (Vanilla) with a current belief over the letters comprising the secret word. ABBEL alternates between updating a belief state given new observations, and selecting an action based solely on the current belief. Thus, ABBEL relies on the ability of a language model to propagate the correct

---

[1]In *Wordle*, the player has 6 tries to guess a 5-letter secret word, receiving feedback about each letter (i.e., whether it is not in the secret, in the secret in a different position, or in the correct position) after each guess.

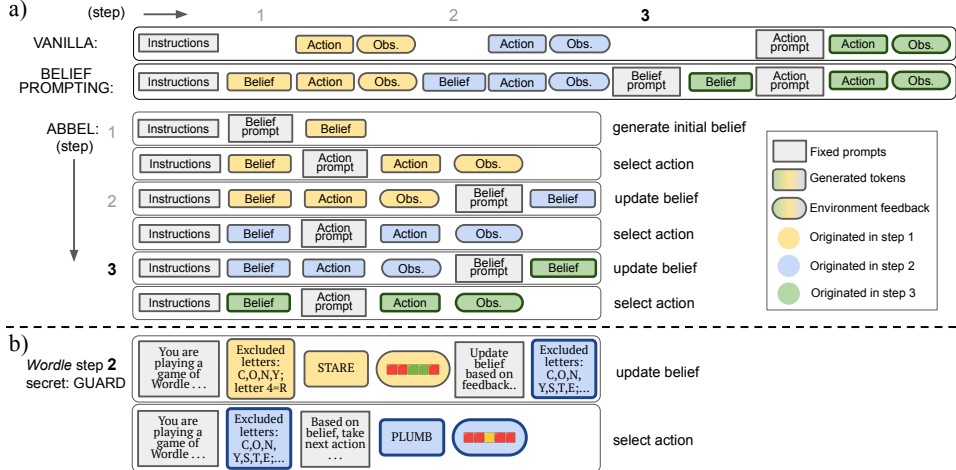

Figure 1: (a) Overview of the belief update and action selection contexts over 3 timesteps under ABBEL, in contrast to the typical multi-step paradigm (VANILLA) or simply prompting for belief generation (BELIEF PROMPTING) which keep all past steps in context. (b) an example step of ABBEL in *Wordle*; actions are word guesses, and observations provide feedback on each letter.

information at each step: the model's output must maintain *sufficient* information for selecting good actions, while discarding superfluous information, e.g., repeated feedback that a letter is not in the secret word, to generate belief states that are *compact* enough to keep the context length manageable in long-horizon settings.

We systematically evaluate current frontier models under ABBEL across six multi-step environments with varying levels of reasoning complexity and structure, and compare to ablations to separately study the effects of prompting for belief generation and removing the interaction history. We find that in many environments, the generated belief states are human-understandable and significantly shorter than the full interaction history without significantly impacting performance, and that conditioning on self-generated beliefs also reduces unnecessary reasoning. While interaction history grows linearly with interaction steps, the lengths of ABBEL-generated beliefs grow much more slowly, even decreasing in some environments as the beliefs concentrate around the answer. However, for each model, we find environments where reduced context decreases task performance, and identify several key causes: propagating erroneous beliefs across steps, hallucinating false memories of previous steps, and repeating uninformative actions because the belief doesn't change without new information.

Considering the significant divergence between ABBEL and typical LLM training settings, we propose to use RL to fine-tune LLM agents to better generate and reason through belief state bottlenecks under ABBEL. In addition to outcome rewards, we introduce belief grading and belief length penalty rewards to train the generation of more accurate and more concise beliefs, respectively. Training Qwen2.5-7B-Instruct with belief grading in a simplified version of *Wordle*, we find ABBEL exceeds the performance of the full-context setting by about 20% while maintaining near-constant-length beliefs. We train ABBEL with a belief length penalty in a multi-objective question-answering setting with much lengthier observations and more extreme horizon generalization from Zhou et al. (2025b), obtaining significantly higher task performance with lower memory usage than MEM1 (Zhou et al., 2025b). Ablating the belief length penalty, we find it only slightly decreased performance, demonstrating that the isolated belief state provides the flexibility to effectively trade-off performance for memory usage without degrading reasoning. We finally study our approach in a more complex environment, ColBench (Zhou et al., 2025a), a collaborative programming setting where the agent must assist a user in writing code through asking for clarifications about the desired behavior. We find that belief grading allows more data-efficient training, and ABBEL learns to perform close to the full-context model while using half as much memory.

## 2 RELATED WORK

**Long context management.** Several recent systems have developed practical solutions for managing long contexts. Context compression methods generate dense representations that, while computationally efficient, sacrifice human-understandability (Chevalier et al., 2023; Jiang et al., 2024). Wang et al. (2025b), Örwall (2024) and Starace et al. (2025) hand-design summarization prompts and pruning strategies specific to their target environments, which requires expert human knowledge of what information must be maintained for each task rather than allowing the agent to learn what to remember as part of its decision-making strategy. Packer et al. (2024) and Xu et al. (2025) process long contexts into an external memory store for the agents to query, which is is an orthogonal approach with different constraints, and can be combined with ABBEL. Wang et al. (2025a) and Yu et al. (2025) recursively update a natural language summary similar to ABBEL's belief state, but they summarize pre-existing long contexts divided into chunks, with no method to update summaries after taking actions. We study the more general multi-step setting where the agent must continually update a summary while actively exploring, which requires reasoning over the summary to select actions that gather missing information needed for the task.

**Multi-step exploration with beliefs.** Various works have studied compact representations of interaction history for multi-step tasks that involve active information-gathering. Kim et al. (2025) improve action selection by first prompting LLMs to explicitly generate beliefs of the current state relative to the goal, though they still include the full interaction history in context. Hard-coded summary statistics of past observations have proven effective for bandit problems (Krishnamurthy et al., 2024; Nie et al., 2025), but lack the flexibility needed for more complex environments. Arumugam & Griffiths (2025) show that frontier models can be effective at belief updating, but they initialize the agents with hand-crafted prior beliefs tailored to each environment, whereas in realistic settings such priors are often unavailable, and they use the suboptimal posterior sampling algorithm to select actions rather than training agents to explore optimally from beliefs. MEM1 (Zhou et al., 2025b) trains LLMs to maintain an internal state, similar to ABBEL's belief state, that summarizes key information during multi-step interaction. However, while ABBEL first generates a belief and then reasons with the belief to select an action, MEM1 directly reasons to select an action and treats the entire reasoning trace as the new internal state. Entangling the beliefs about the task with the reasoning harms conciseness and interpretability, and makes it difficult to steer or compress the beliefs during training in contrast to ABBEL's isolated belief state.

## 3 FORMULATION

**Problem Setup.** We model each environment as a Partially Observable Markov Decision Process, using *Wordle* as an example environment for grounding our formulation. In *Wordle*, the objective is to identify a secret 5-letter word in fewer than 7 turns by guessing a 5-letter word at each step. Each *task* corresponds to a randomly sampled hidden initial state $s_0$, e.g., (secret:GUARD, step:0). At each step the agent selects an action $a_t$ from the action space, e.g., 5-letter English words. The hidden state $s_{t+1}$ is updated based on $s_t$ and $a_t$, which in *Wordle* simply increments the step counter. The agent receives reward $r_t$ and observation $o_t$ both conditioned on $a_t$ and $s_t$, e.g., $r_t = 1$ if $a_t =$ GUARD and step $< 7$ otherwise $r_t = 0$, and $o_t$ is feedback on each letter in $a_t$ (i.e., whether the letter is not present in the secret word, present at a different position, or present at the guessed position) and the new step count (see Fig. 1).

**Belief Bottleneck Interaction Framework.** We use LLMs to implement context-conditioned policies $a_t \sim \pi(\cdot \mid c_t)$. In the typical multi-step paradigm, the context includes the full interaction history of observations and actions $h_t = \langle a_1, o_2, a_2, o_3, \ldots, a_{t-1}, o_{t-1} \rangle$, as shown in Fig. 1 (*Vanilla*), while in ABBEL it contains only a current belief. In ABBEL, the agent is called twice at each step $t$: first, conditioned on the environment instructions $p_I$ (e.g., how to play *Wordle*) and the last belief, action, and observation, and belief prompt $p_b$, we generate a new belief $b_t \sim \pi(\cdot \mid p_I, b_{t-1}, a_{t-1}, o_{t-1}, p_b)$ (*Update belief* in Fig. 1). Next, all steps before $t$ are removed from the context, and $\pi$ is called with action prompt $p_a$ and the newest belief $b_t$ to select the next action $a_t \sim \pi(\cdot \mid p_I, b_t, p_a)$ (*Select action* in Fig. 1), resulting in a new observation $o_t$ from the environment. See Appendix B for the full details. We measure the performance of $\pi$ in each environment by its expected performance across the task distribution, e.g., the uniform distribution over all possible 5-letter secret words.

## 4 EVALUATING FRONTIER MODELS WITH BELIEF BOTTLENECKS

We investigate to what extent current frontier models can already generate and reason through natural language belief states as bottlenecks in reasoning. We use a purely prompting-based approach, following the framework described in section 3.

### 4.1 ENVIRONMENTS

We evaluate across six multi-step environments from Tajwar et al. (2025) spanning various levels of reasoning complexity and structure.[2] *Wordle* and *Mastermind* demand complex reasoning using highly structured feedback on each position of a secret word or 4-digit code. *Mastermind* has the same rules as *Wordle* (described in Section 3), except feedback only reveals the total number of guessed digits in the correct position, or in the code but in a different position. *Twenty Questions* and *Guess My City* involve iteratively narrowing down a search space of topics or cities by asking a sequence of questions. In contrast, both actions and observations in *Murder Mystery* and *Customer Service* are free-form descriptive sentences: actions correspond to clue-gathering or troubleshooting instructions, and observations, generated by GPT-4o-mini, describe what the detective discovers or how the customer responds. The goal is to identify the culprit or correctly diagnose the customer's problem, respectively.

### 4.2 MODELS AND FRAMEWORKS

We evaluate Gemini 2.5 Pro, DeepSeek R1, and DeepSeek V3 using chain-of-thought prompting. For each model, we compare ABBEL with two variations. The first is a standard multi-step interaction framework (Fig. 1, VANILLA) where at each step the agent is prompted with the initial instructions followed by the full interaction history of actions and observations (not including reasoning), and finally a prompt to generate the next action. The second framework (Fig. 1, BELIEF PROMPTING) follows ABBEL in first prompting to update beliefs and then prompting to select an action given the beliefs at each step, but the full interaction history remains in context, ablating the information bottleneck aspect of ABBEL. We sample 40 task instances from each environment and report the mean and standard error of the mean of task outcomes (Success Rate).

### 4.3 RESULTS

**Task Performance.** We first analyze how well frontier models perform under each framework. Fig. 2a presents the success rates for each setting. We find that Gemini 2.5 Pro with ABBEL maintains or even exceeds the performance of both full-context settings in most tasks. However, the Deepseek models generally perform worse under all frameworks and show greater drops in performance under ABBEL, with the exception of *Twenty Questions*. We then examine the performance of BELIEF PROMPTING to separately study the effects of prompting for belief generation, and acting on the belief state bottleneck. Here, a belief state is maintained, but in contrast to ABBEL, we condition action generation both on the belief state and the full history. Prior work has found that conditioning on interaction history alongside a belief summary is helpful for long sequential decision-making tasks (Kim et al., 2025). In our experiments, we find that BELIEF PROMPTING rarely outperforms VANILLA and sometimes substantially decreases performance. Secondly, we investigate belief *sufficiency*, comparing ABBEL and BELIEF PROMPTING. We observe that the weaker Deepseek models generally struggle more with generating sufficient beliefs in environments with low information structure (*Customer Service* and *Murder Mystery*), where it is more ambiguous what information should be maintained in the beliefs. Even Gemini 2.5 Pro fails to generate sufficient beliefs across all environments, as evidenced by the small performance drop in *Mastermind*.[3]

**Belief State Compactness and Interpretability.** We next investigate if ABBEL can reduce the context length for frontier models by examining the compactness of belief states generated through ABBEL across different models and tasks, shown in Fig. 2b. In most cases, beyond the first few steps, the belief states were significantly shorter than the length of the interaction history (gray dashed

---

[2]Table 3 in the Appendix summarizes key characteristics of each environment.

[3]Surprisingly, Gemini 2.5 Pro performs much better under ABBEL than VANILLA in *Murder Mystery*. We find that VANILLA is more biased to keep gathering clues and run out of steps before making an accusation.

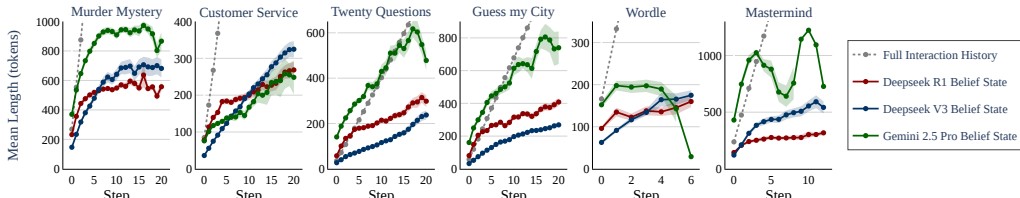

(a) Performance of frontier models across 6 environments under the typical multi-step paradigm (VANILLA), prompting for beliefs before acting (BELIEF PROMPTING), and ABBEL. Error bars indicate standard error of the mean. In most tasks, Gemini 2.5 Pro maintained performance with ABBEL despite significantly reduced context lengths.

(b) Average length of beliefs generated under ABBEL compared to full interaction histories. While history grows linearly over interaction steps, the belief lengths generally grow more slowly and are significantly shorter after the first few steps.

Figure 2: Behavior of frontier models across environments and frameworks.

line). While the history always grows linearly with the number of interaction steps, belief lengths grow more slowly, plateauing or even decreasing in some environments as possibilities were ruled out, with the exception of Gemini 2.5 Pro in *Twenty Questions* and *Guess My City*. By inspection we found that all models generated human-understandable beliefs, which allowed us to better understand model behavior. For instance, in *Twenty Questions* we find that Gemini 2.5 Pro concatenates all information from the observations, which explains why the length grows linearly with time on par with the history, whereas DeepSeek R1 maintains a compact description of the posterior beliefs (see Appendix C for examples).

**Impact on Reasoning.** Finally, we investigate how ABBEL affects reasoning for action selection, where models are prompted to think step-by-step before choosing an action, conditioned on some context. We find that conditioning on belief states generated by ABBEL and BELIEF PROMPTING rather than full histories significantly reduces reasoning length for comparable performance in several environments (see Figure 6). We also find ABBEL often uses even less reasoning than BELIEF PROMPTING while achieving similar success rates. Thus, using belief states as a bottleneck provides an additional benefit of preventing unnecessary extra reasoning over interaction histories when beliefs are sufficient. See Appendix D for more analysis.

We additionally inspect the traces to get further insight into the challenges of reasoning through a belief bottleneck. We find that performance of ABBEL is impacted when the agent does not update the belief state after uninformative observations (e.g., in *Customer Service* when the customer responds "I'm not sure" to the agent's question), causing it to take the same action again, whereas if the action selection step is conditioned on the interaction history (including previous actions), it is much less likely to repeat an uninformative action. Additionally, in environments requiring more complex reasoning (*Wordle* and *Mastermind*), we find many cases where belief state errors are introduced and propagated from one step to the next. If errors are propagated, models have the opportunity to self-correct the belief state if they receive contradictory observations, but the wasted turns may be irrecoverable; whereas access to the full history enables earlier error detection and perfect posterior reconstruction. We find two main causes of belief state errors: incorrectly updating on the new observation due to mistakes in reasoning (e.g., falsely assuming that the secret code cannot contain repeated characters), and hallucinating false memories of past interactions (see Appendix D.1 for an example).

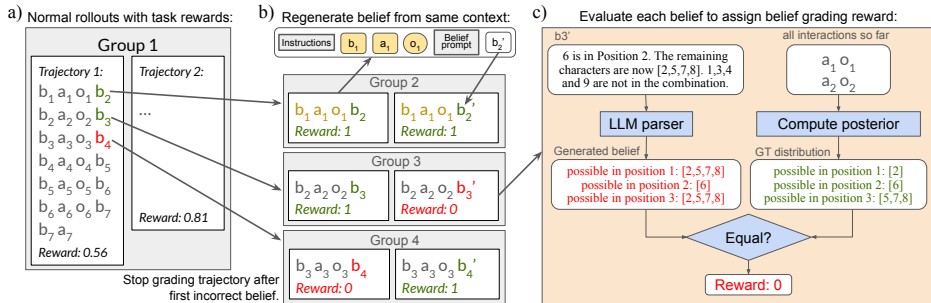

Figure 3: Overview of belief grading. Beliefs, actions and observations generated at timestep $t$ are denoted by $b_t$, $a_t$, and $o_t$, respectively. After collecting trajectories from the current ABBEL policy (a), each step is copied into a new group consisting of the original belief update and a newly generated posterior belief from the same context (b), which are each assigned rewards by a belief grader (c). The grader shown here was customized for *Combination Lock*, a 3-digit version of *Wordle*. The policy is finally updated with GRPO using both the trajectory groups and the belief groups.

# 5 REINFORCEMENT LEARNING TO ACT THROUGH BELIEF BOTTLENECKS

We found in Section 4 that ABBEL can already lead to significantly shorter yet interpretable contexts for frontier models, and belief bottlenecks also have potential for improving reasoning efficiency. However, for each frontier model we found environments where there was still significant room for improvement, in either the task performance or the conciseness of the beliefs. Reinforcement learning (RL) has been shown to improve general abilities across task structures and input distribution shifts compared to SFT alone (Nie et al., 2025; Kirk et al., 2024; Tajwar et al., 2025). We propose to use RL to improve LLMs' abilities to generate and reason through belief bottlenecks under ABBEL.

## 5.1 METHODS

RL with outcome-based rewards naturally incentivizes learning to accurately maintain the relevant information in the beliefs for completing the task, without requiring task-specific knowledge or demonstrations. In addition, we experiment with rewards that leverage ABBEL's isolated belief states to provide additional training signal.

### 5.1.1 BELIEF LENGTH PENALTIES

For settings where ABBEL generates bloated belief states, we propose to add a small negative reward penalizing the token length of the belief states. Because ABBEL's belief states are separated from the reasoning, this penalty encourages more concise beliefs without degrading reasoning capabilities. The penalty for a trajectory is proportional to the length of longest belief state in the trajectory, and like Arora & Zanette (2025), we apply it after advantage normalization, to reduce its impact as beliefs get shorter to avoid over-compression. See Appendix G.3 for details.

### 5.1.2 BELIEF GRADING

In environments requiring more complex belief update reasoning such as *Wordle* and *Mastermind*, it may be difficult to learn to generate accurate beliefs from a sparse outcome reward. Inspired by the use of belief grading for tuning context summarization prompts in software engineering tasks (Wang et al., 2025b), we propose to add shaping rewards based on the quality of the generated beliefs.

Adding rewards for every belief state directly to each trajectory's outcome rewards may cause reward hacking, as rewards could be maximized by solving the task less efficiently to collect more step-wise belief rewards (Lidayan et al., 2025). Instead, we treat belief generation as a separate task, creating additional "trajectories" consisting of single belief update generations to which we assign the grading rewards (Fig. 3a and 3b). To do this, we collect belief states generated during the environment's multi-step roll out, and for each belief we prompt ABBEL again to generate another belief from the same context to create a size-2 group for GRPO (Fig. 3b). We then grade and assign rewards to each

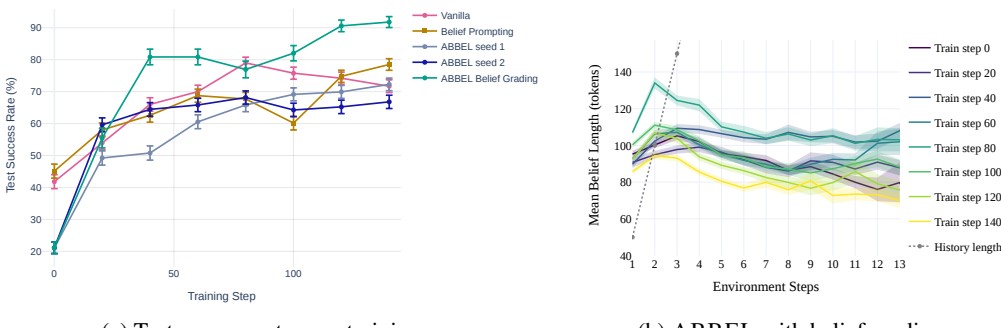

(a) Test success rate over training.    (b) ABBEL with belief grading.

Figure 4: Test behavior of Qwen2.5-7B-Instruct trained in *Combination Lock*. (a) Task success rates over training steps show ABBEL quickly reduces its performance gap with other frameworks, while Belief Grading allows ABBEL to surpass the other frameworks. (b) The beliefs generated by ABBEL BELIEF GRADING initially increase in length but eventually become even shorter over training, and remain significantly shorter than the interaction history beyond the first 2 environment steps.

belief (Fig. 3c), which provides a learning signal whenever the two beliefs in a group receive different grades. The policy gradient step for belief grades and outcome-based rewards is applied concurrently. See Algorithm 2 for more details. Different grading functions may be used for different environments; here we show an example for *Combination Lock*, a 3-digit version of *Wordle* (details in Section 5.2.1). We also propose a domain-general heuristic that does not require parsing or ground-truth posteriors in Section 5.2.3.

## 5.2 EXPERIMENTS

To evaluate our approach, we train ABBEL in *Combination Lock*, which requires complex belief update reasoning, multi-objective QA (Zhou et al., 2025b), with much lengthier 300-word observations and extreme horizon generalization (from 2 questions and 6 steps to 16 questions and 20 steps), and *ColBench* (Zhou et al., 2025a), a more complex collaborative coding setting. In all experiments, we train Qwen2.5-7B-Instruct with chain-of-thought prompting, and use GRPO in VeRL-agent (Feng et al., 2025), a multi-context synchronous rollout framework (for full details see Appendix G).

### 5.2.1 COMBINATION LOCK

**Environment and Metrics.** *Combination Lock* is a 3-character version of *Wordle* proposed by Arumugam & Griffiths (2025); we train with a vocabulary of 10 digits and 12-step horizon, and test on a disjoint vocabulary of 16 letters and 16-step horizon. Each episode ends with reward $(H + 1 - \text{steps to find code})/H$ if the code was identified, and $-1$ otherwise. As a coarse-grained measure of performance, we report the fraction of episodes ending in identifying the secret code (Success Rate). To cleanly quantify exploration efficiency, we also measure the "Cumulative Regret" over each trajectory, which increases by 1 at every step that the code has not been identified such that the mean Cumulative Regret at step $H$ is the mean number of steps taken to find the code.

**Experimental Setup.** As *Combination Lock* involves complex belief update reasoning, we train ABBEL with belief grading as outlined in Section 5.1.2 (ABBEL BELIEF GRADING). We also train without belief grading (ABBEL), as well as the full-context BELIEF PROMPTING and VANILLA settings described in Fig.1.

**Belief Grader.** In *Combination Lock* it is possible to compute the ground truth posterior exactly from the previous actions and observations in the trajectory. To grade each belief in *Combination Lock*, we first used Grok-4-Fast-Free to parse it into a list of possible numbers at each position, which we compared to the ground truth posterior, generating a reward of 1 when they were identical and 0 otherwise (Fig. 3c). We stop grading each trajectory after the first step with an incorrect belief, to avoid penalizing beliefs that were only incorrect due to propagating errors from the previous step.

**Results.** In line with our findings from Section 4.3, the initial performance of all ABBEL agents was significantly lower than either the baseline (VANILLA) or BELIEF PROMPTING (Fig. 4a). However, we find that RL with belief grading is highly effective, resulting in ABBEL-BELIEF GRADING outperforming both. The beliefs remain concise; in fact, we find that they first increase in length, but then decrease later in training (Fig. 4b), which could be a side-effect of the grading encouraging the model to generate beliefs that are easier to parse with an LLM (see Appendix E for examples). Ablating the belief grading, we find it played a major role in boosting performance, especially Cumulative Regret (Fig. 10a), though we note that RL is still effective for ABBEL, leading to its success rate quickly increasing to bridge the gap with the full-context models. We also find that ABBEL without grading learns to generate longer belief states over training (though still significantly shorter than the full interaction history past the first two steps (Fig. 10b).

### 5.2.2 MULTI-OBJECTIVE QUESTION ANSWERING

**Environment and Metrics.** In the multi-objective question answering (QA) environment introduced by Zhou et al. (2025b) each task requires the agent to answer a set of questions (objectives), by iteratively querying an external knowledge base before generating a final answer composed of semicolon-delimited answers to each question. Each query retrieves the first 100 words of the three most relevant documents in the knowledge base. During training, each task involves only 2 questions and a horizon of 6 steps, while we evaluate on tasks with up to 16 objectives and 20 steps. We use the Exact Match Count (EM), defined as the number of answers that exactly match the correct answer text, as both the reward and performance metric. We measure memory efficiency with the Peak Token Usage metric proposed by Zhou et al. (2025b), which is the maximum sequence length (input and output, excluding the system prompt) over all steps in each trajectory. We report mean and standard error over the test set for each metric.

**Experimental Setup.** As this environment involves very lengthy observations, we experiment with training ABBEL with a belief length penalty (ABBEL-LP) to further decrease memory usage. We also train with no penalty (ABBEL) and evaluate with no RL at all (ABBEL Zero). We compare with MEM1 (Zhou et al., 2025b), which also uses RL to train LLMs to generate and act on context summaries instead of full interaction histories. However, rather than generating a separate belief state, the entire reasoning trace is used as the memory that gets carried forward to the next step. We refer to both this memory and ABBEL's belief states as *internal states*. We compare with the metrics reported by Zhou et al. (2025b) for MEM1 (MEM1 Base, trained from Qwen2.5-7B-Base), and an untrained Qwen2.5-14B-Instruct model operating in the full context setting (VANILLA 14B Zero-Shot). We also re-implement MEM1 by training a Qwen2.5-7B-Instruct model under MEM1's prompting and rollout framework (MEM1 Instruct)[4] for an apples-to-apples comparison with ABBEL. As a measure of best-case performance, we train a Qwen2.5-7B-Instruct model in the full-context setting (VANILLA) and also evaluate its zero-shot performance (VANILLA Zero).

**Results.** ABBEL achieves significantly higher performance than all other memory models for more than 2 objectives (Fig. 5a). Inspecting the belief states, we find that they remain concise and interpretable, summarizing what is known so far about the answers to the questions. Meanwhile MEM1's internal states are significantly longer (Fig. 5b), containing reasoning for drawing conclusions from previous search results (see Appendix F for examples). Though ABBEL's shorter internal state doesn't make a big difference to Peak Token Usage relative to MEM1 due to the length of the reasoning and environment feedback, the more concise beliefs may help performance by being easier to reason over. The lower performance of ABBEL Zero-Shot confirms that RL was effective, while causing only a small increase to memory usage. The belief length penalty further shrinks the belief states, making ABBEL LP significantly more memory-efficient than MEM1 (Fig. 5c), with only a slight decrease in performance compared to ABBEL while still significantly outperforming MEM1. Inspecting the beliefs, we observe that they remain interpretable yet more concise (see Appendix F for examples). This shows that ABBEL provides the flexibility to efficiently trade-off memory usage for performance. Note that this shaping reward cannot be applied to MEM1 as it does not isolate the belief from the reasoning in the internal state, so such a reward would have the adverse effect of penalizing reasoning. The trained VANILLA model only performs slightly better than ABBEL, with no advantage at the 16 objective setting despite access to the full context and using 9.5x as much memory. In addition, both zero-shot VANILLA models cannot handle 16 objectives at all (scoring

---

[4]In our experiments we found training from Qwen2.5-7B-Instruct outperformed training from the base model.

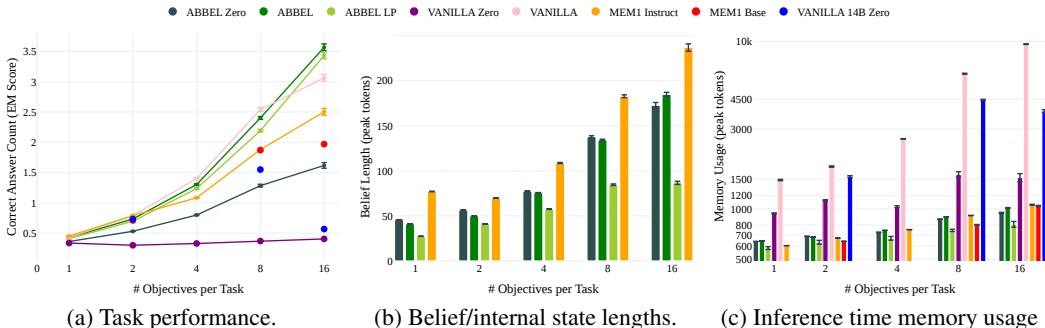

(a) Task performance.   (b) Belief/internal state lengths.   (c) Inference time memory usage

Figure 5: Model comparison in multi-objective QA. ABBEL performs closest to the full-context VANILLA model for 4+ objectives, and training with a length penalty on the belief state (ABBEL LP) remains competitive while using much less memory.

Table 1: Multi-objective QA results. Arrows indicate desired directions. Results for MEM1 Base and VANILLA 14B Zero from Zhou et al. (2025b). Memory models listed in the bottom section.

| Model | 2-Objective | | 8-Objective | | 16-Objective | |
|---|---|---|---|---|---|---|
| | EM Score ↑ | Tokens ($\times 10^2$) ↓ | EM Score ↑ | Tokens ($\times 10^2$) ↓ | EM Score ↑ | Tokens ($\times 10^2$) ↓ |
| VANILLA Zero | 0.30 | **11.25±0.09** | 0.37 | **16.06±0.59** | 0.40 | **15.40±0.84** |
| VANILLA 14B Zero | 0.73 | 15.60±0.19 | 1.55 | 44.70 ± 0.37 | 0.57 | 38.40±0.71 |
| **VANILLA** | **0.79** | 17.87±0.15 | **2.54** | 64.07±0.44 | **3.06** | 96.08±0.37 |
| MEM1 Base | 0.71 | **6.40±0.02** | 1.87 | 8.01±0.06 | 1.97 | 10.40±0.09 |
| MEM1 Instruct | **0.79** | 6.69±0.01 | 1.88 | 9.13±0.03 | 2.50 | 10.58±0.07 |
| ABBEL Zero | 0.53 | 6.85±0.01 | 1.28 | 8.67±0.04 | 1.62 | 9.46±0.08 |
| ABBEL | 0.73 | 6.78±0.01 | **2.40** | 8.95±0.03 | **3.57** | 10.12±0.06 |
| **ABBEL LP** | 0.70 | 6.56±0.01 | 2.19 | **7.61±0.02** | 3.43 | **7.64±0.04** |

about 3.5x lower than ABBEL Zero), suggesting that this setting may be approaching the limit of what long-context models can handle.

### 5.2.3 COLLABORATIVE PROGRAMMING

**Environment and Metrics.** We use the collaborative back-end programming environment from the ColBench benchmark introduced by Zhou et al. (2025a), where the agent must collaborate with the user to write a Python function of up to 50 lines. The agent is initially provided with an under-specified high level description and the function signature, and can ask the user up to 10 questions to gather information before finally submitting code. The generated code is finally evaluated by 10 hidden unit tests, yielding an outcome reward equal to the fraction of unit tests passed. We report the mean fraction of passing unit tests (Test Pass Rate), and the fraction of tasks with all 10 tests passing (Success Rate). As in Section 5.2.2, we measure Peak Tokens to evaluate memory usage. The human user is simulated by Gemma 3 27B-it with access to the hidden tests and a reference solution, prompted to behave like a human that needs help.

**Experimental Setup and Belief Grader.** We train 2 seeds each of ABBEL with and without belief grading (BG), and one seed in the full-context setting described in Fig.1 (VANILLA), evaluating after 0, 50 and 100 training steps. Ground-truth posteriors are unavailable in ColBench, so we use a fully domain-general belief grader: how useful the generated belief $b_{t+1}$ is for reconstructing the most recent observation $o_t$ given previous belief $b_t$ and action $a_t$, to encourage $b_{t+1}$ to integrate information in $o_t$ that isn't already in $b_t$. We define this as the mean log probability under the agent model of the tokens in the last observation conditioned on $b_{t+1}$, $b_t$, and $a_t$, i.e.,

$$f_{\text{BG}}(b_{t+1}) = \frac{1}{|o_t|} \log p(o_t \mid b_t, a_t, b_{t+1}). \tag{1}$$

This expression is proportional to $\log p(b_{t+1} \mid b_t, a_t, o_t) - \log p(b_{t+1} \mid b_t, a_t)$ plus a constant (by application of Bayes' rule), where the second term encourages $b_{t+1}$ to contain new information relative to the prior, while the first term encourages that new information to be explainable by $o_t$.

**Results.** We find zero-shot ABBEL and VANILLA are on-par due to ABBEL biasing the agent to ask more questions before submitting code (an average of about 6, versus only 2.8 for VANILLA), making it more likely to get all necessary clarifications. This also explains why zero-shot VANILLA has low Peak Tokens. We again find that ABBEL's performance improves with RL while remaining far more memory-efficient than VANILLA: ABBEL's step 100 performance is only 11.5% lower than VANILLA while using 49% as much memory. We observe that belief grading helps ABBEL learn to add more useful information to its beliefs, as at step 50 ABBEL-BG's belief states were on average 24% longer and its performance was significantly better compared to ABBEL (see Appendix G.4 for examples) and on-par with VANILLA while using less than half as much memory. ABBEL without belief grading learns more slowly, only catching up at step 100.

Table 2: Model comparison on ColBench. Arrows indicate desired directions. We report the mean and SEM over 2 seeds for ABBEL and ABBEL-BG, and over the test set of 1 seed for VANILLA.

| Step | Model | Test Pass Rate ↑ | Success Rate ↑ | Peak Tokens ($\times 10^2$) ↓ |
|------|-------|------------------|----------------|------------------------------|
| 0 | VANILLA | **0.2827±0.0125** | **0.1748±0.0119** | 4.5938±0.1532 |
|   | ABBEL | 0.2642±0.0125 | **0.1709±0.0118** | **3.2953±0.0525** |
| 50 | VANILLA | **0.4456±0.0139** | **0.3047±0.0144** | 8.9805±0.1396 |
|   | ABBEL | 0.3844±0.0140 | 0.2651±0.0093 | **3.4078±0.0499** |
|   | ABBEL-BG | **0.4560±0.0132** | **0.3228±0.0079** | 3.9693±0.2542 |
| 100 | VANILLA | **0.5260±0.0141** | **0.3936±0.0153** | 7.8845±0.1084 |
|   | ABBEL | 0.4655±0.0112 | 0.3286±0.0121 | 3.8614±0.0711 |
|   | ABBEL-BG | 0.4577±0.0004 | 0.3262±0.0021 | **3.4149±0.3210** |

## 6 DISCUSSION

We introduce ABBEL, a general framework for LLM agents to maintain manageable and interpretable contexts for long horizon interactive tasks via generating natural language beliefs. ABBEL provides a valuable testbed for exploring the limitations of models in constructing beliefs, and opens up myriad possibilities for supervision and controllability during training.

Evaluating frontier models in ABBEL across diverse multi-step environments, we find that they maintain interpretable beliefs that are significantly shorter than full interaction histories, and the bottleneck can reduce unnecessary reasoning. However, we find the models fail to generate both concise and sufficient belief states in all environments, with failure modes including propagating belief errors across steps and hallucinating false memories of previous steps. We thus propose RL in ABBEL as a general method for post-training LLM agents to more effectively generate and reason through beliefs, and introduce additional methods for steering RL through belief bottlenecks. In particular, we propose *belief length penalties* to generate more concise beliefs without degrading reasoning, and *belief grading* to reward the generation of high quality beliefs. In *Combination Lock* we show that RL with a task-specific belief grader allows ABBEL to outperform models with full history access. In multi-objective QA we show that ABBEL outperforms contemporaneous approaches for multi-step context management, with belief length penalties allowing ABBEL to efficiently trade off performance and memory use. Finally, we demonstrate ABBEL is also effective in the more complex ColBench environment, with a domain-general belief grading heuristic helping ABBEL learn to integrate more useful information into its beliefs.

In our work we focus on the improvements which can be gained by improved belief generation, but for practical settings this may be combined with additional external memory tools such as Packer et al. (2024) for even better results. Additionally, though ABBEL updates the belief state after every action, in practice beliefs may be updated much less frequently for lower computational costs. Though we study multi-step settings, recent work suggests methods like ABBEL may also be helpful for single-step long reasoning problems by formulating beliefs over internal reasoning, and treating chunks of reasoning as observations to update on (Yan et al., 2025).

# 7 REPRODUCIBILITY STATEMENT

We have provided the full prompts used in Appendix B and the RL training details including the hyper-parameters used in Appendix G. We have also open-sourced our code in an anonymous repository available here. We believe that with our code and prompts, all results from the paper should be completely reproducible.

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

## A    LLM USAGE IN PAPER WRITING.

LLM tools were used minimally for finding related work, polishing writing, e.g., rephrasing sentences to flow more naturally, and editing code to reformat figures.

## B    BELIEF BOTTLENECK ROLLOUT

---
**Algorithm 1** Belief Bottleneck Rollout
---
**Require:** Instructions $p_I$; horizon $H \in \mathbb{N}$; step function $T : \mathcal{S} \times \mathcal{A} \to \mathcal{S} \times \mathcal{O}$; initial state $s_0$.
**Require:** Belief generation prompt $p_b$; action selection prompt $p_a$; policy $\pi$.
  $t \leftarrow 0$
  $s \leftarrow s_0$
  $b \leftarrow$ "This is the start of the game. No beliefs right now."
  **while** $t \leq H$ **do**
    $a \sim \pi(\cdot | p_I, b, p_a)$                         ▷ Action selection
    $s, o \leftarrow T(s, a)$               ▷ Environment step: get new state and observation.
    $t \leftarrow t + 1$
    $b \sim \pi(\cdot | p_I, b, a, o, p_b)$                    ▷ Belief update
  **end while**
---

See Algorithm 1 for the rollout algorithm. Below, we show one sample and associated prompts for the game *Wordle* with Deepseek R1 (illustrated in Fig. 1).

**Game State**

**Secret:** GUARD
**Step:** 2 out of 6
**Goal Reached:** No

**Instructions**

You are playing a game of Wordle.
Format your response in the following way: <Think> Any step-by-step, short and concise thinking to strategically determine the next guess for the secret word </Think> <Answer> your guess of what the word should be </Answer>
The game begins now, please make your first guess about the secret five-letter word!

**Current Belief State**

Excluded letters: C, O, N, Y
Included letters: R (must be in position 1, 3, 4, or 5)
Target word contains R and excludes C, O, N, Y.

**Action Prompt**

This is the game information:
`<game_instructions>`
You are currently taking your attempt `<step>` out of `<horizon>` attempts. Look at the current belief state and history and give an answer based on it.
Give an answer that leads to optimal exploration and do not be greedy unless it is the last attempt. Try to maximize the amount of information you have so that you can solve the task correctly.
Belief: `<curr_belief>`
Please format your response as: <Answer>your answer in the correct format mentioned </Answer>

**Action and Observation**

**Action:** STARE
**Observation:**
First letter, s, is not in the target word
Second letter, t, is not in the target word
Third letter, a, is correct and in the correct position in the target word
Fourth letter, r, is correct and in the correct position in the target word
Fifth letter, e, is not in the target word

**Belief Update Prompt**

This is the game information:
`<game_instructions>`
Look at the current belief and the agent's action and environment response on that belief. You have to update the current belief based on the action and response, while maintaining important information about the game state needed to take optimal future actions.
Current belief: `<curr_belief>`
Agent's action: `<action>`
Environment's response: `<observation>`
Output the updated belief state inside <BELIEF> and </BELIEF> tags.
Understand that only the generated belief is fed to the agent to pick the next action, not the history, so be sure to include all necessary information.

**Updated Belief State**

Excluded letters: C, O, N, Y, S, T, E
Included letters: A (position 3), R (position 4)
Target word contains A and R, excludes the listed letters, and has A in position 3 and R in position 4.

Table 3: Characteristics of frontier model evaluation environments.

| Environment | Horizon | Complex Reasoning | Information Structure | Answer Space Size |
|---|---|---|---|---|
| Murder Mystery | 20 | No | Low | 3 (suspects) |
| Customer Service | 20 | No | Low | $\sim 10^2$ (faulty parts) |
| Twenty Questions | 20 | No | Medium | $\sim 10^3$ (e.g., animals) |
| Guess My City | 20 | No | Medium | $\sim 10^3$ (all cities) |
| Wordle | 6 | Yes | High | 2315 (5-letter words) |
| Mastermind | 12 | Yes | High | $10^4$ (4-digit numbers) |

## C  FRONTIER MODEL-GENERATED BELIEF STATE EXAMPLES

We compare examples of the belief states generated by DeepSeek R1 and Gemini 2.5 Pro operating under ABBEL. Each are at step 15 of the *Twenty Questions* game, where the answer is "waistcocat".

**DeepSeek R1**

The garment is a standalone structured, strapless, sleeveless formal bridal piece, not layered over other garments (confirmed). Excludes outerwear (boleros, jackets), full gowns, corsets, bustiers, and undergarments. Focus narrows to standalone bridal separates with stiffened fabrics (taffeta, satin) or layered construction for structural elegance, lacking boning. Key possibilities: embellished strapless bridal tops (beaded/embroidered), corsage-inspired designs with tailored support, or reinforced bridal camisoles/decorative tulle overlays. Must function independently (paired with skirts) without waist-flared elements or basque terminology. Core attributes: bridal formality, structural integrity via tailoring or stiff fabrics, no corsetry associations.

**Gemini 2.5 Pro**

I am playing a game of 20 Questions to identify a piece of clothing. I have now used 15 out of 20 questions.

Here's what I know so far:

- **Fact 1:** The item is worn on the upper body.
- **Fact 2:** It is NOT typically worn as outerwear.
- **Fact 3:** It typically has a collar.
- **Fact 4:** It typically has buttons all the way down the front.
- **Fact 5:** It is typically worn by men.
- **Fact 6:** It is typically worn for formal or business occasions.
- **Fact 7:** It has short sleeves.
- **Fact 8:** It features decorative vertical pleats or embroidery on the front.
- **Fact 9:** It is NOT a Guayabera shirt.
- **Fact 10:** It is NOT considered a form of national dress for a specific country.

- **Fact 11:** It is specifically worn with a tuxedo.
- **Fact 12:** It is NOT called a "short-sleeved tuxedo shirt".
- **Fact 13:** Its decorative front does NOT feature ruffles.
- **Fact 14:** Its decorative front does NOT consist of a distinct, separate panel of fabric (a "bib").
- **Fact 15:** The decorative vertical pleats on the front are NOT known as "pin-tucks".

Based on the cumulative information, my confidence has been updated.

- **Analysis:** The "No" answer to my last question, "Are the decorative vertical pleats on the front known as 'pin-tucks'?" is a major setback. My leading hypothesis of a "pleated-front shirt" or "pin-tuck shirt" has been invalidated. This means that if the decorative elements are pleats, they are a very obscure type.
- **Re-evaluation:** I must now reconsider the other half of Fact 8: "...pleats **or** embroidery". Having exhausted the most common types of pleated fronts (ruffles, bib, pin-tucks), it is highly likely that the decorative elements are not pleats at all, but rather a form of embroidery. The object remains a very unusual item: a short-sleeved formal shirt worn with a tuxedo, featuring vertical decorations that are not ruffles, not a bib, and not pin-tucks.
- **Current Prime Candidate:** There is no longer a single prime candidate for the item's name. The identity is very mysterious. The immediate goal is to confirm the nature of the decoration. My new leading hypothesis is that the item is a shirt with vertical **embroidery** on the front.

My confidence in any specific name is now low. However, I am confident that the next step must be to pivot away from pleats and investigate the "embroidery" possibility directly.

## D  FRONTIER MODEL REASONING ANALYSIS

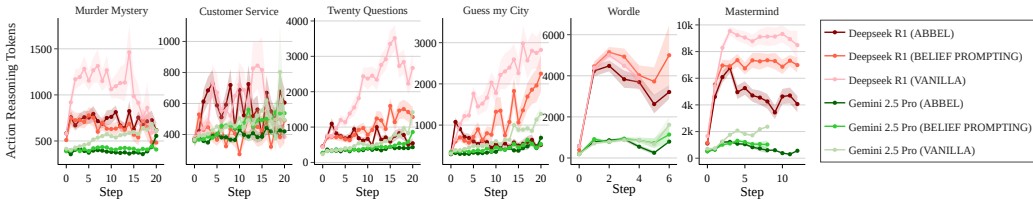

Figure 6: Lengths of reasoning traces for action selection across steps. Some models have no data at higher steps because all episodes ended early. DeepSeek V3 is not shown because it is not a reasoning model. Access to prior beliefs reduces reasoning in most environments, while ABBEL reduces reasoning even more than belief prompting alone.

Figure 6 shows the average length of reasoning used for action selection for DeepSeek-R1 and Gemini-2.5-Pro.[5] Conditioning on belief states generated by ABBEL and BELIEF PROMPTING rather than full histories significantly reduces reasoning length for comparable performance in several environments. We find that this is because the reasoning models naturally integrate information from the interaction history as the first step of reasoning, and access to beliefs allows them to skip this part of the reasoning process. We also find ABBEL often uses even less reasoning than BELIEF PROMPTING while achieving similar success rates (e.g., Deepseek R1 in *Twenty Questions*, *Guess my City* and *Mastermind*). Inspecting the reasoning traces (see Appendix D.2 for examples), we find that

---

[5]Only reasoning summaries, rather than full reasoning traces, were available for Gemini-2.5-Pro. We assume that lengths of reasoning summaries correlate with total reasoning length.

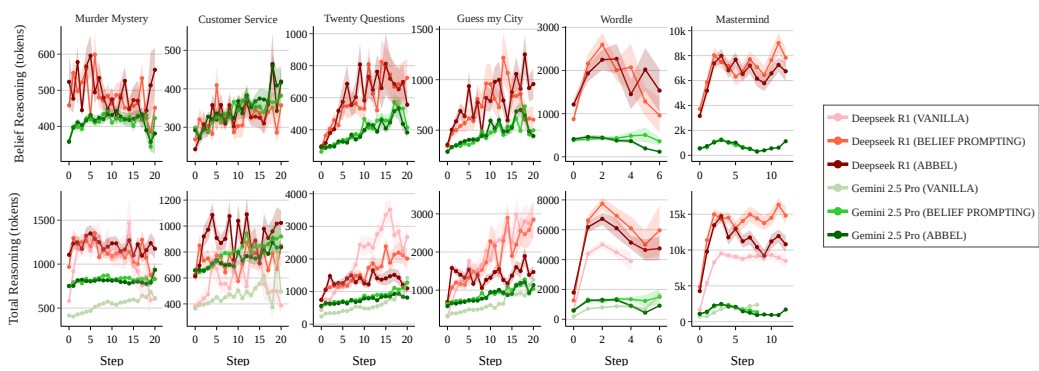

Figure 7: Reasoning trace length for belief generation (top) and the total reasoning length at each step, summing the belief and action selection reasoning lengths (bottom).

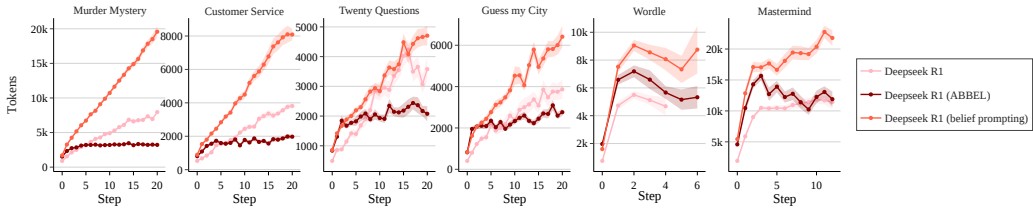

Figure 8: The total number of tokens processed at each step, including both input (i.e., the context) and output (i.e., reasoning, actions and belief states). This remains almost constant for ABBEL, while in many environments it increases nearly linearly for the other frameworks.

R1 has a strong prior to ignore the belief state and reconstruct a posterior from the interaction history when available, so using belief states as a bottleneck provides an additional benefit of preventing unnecessary extra reasoning over histories when beliefs are sufficient. For some environments, even the total length of both action and belief reasoning for R1 with ABBEL was less than VANILLA, with no drop in success rate (see Fig. 7). Accounting for all tokens involved at each interaction step, including input contexts, output reasoning, beliefs, and actions, we find ABBEL uses fewer tokens and requires less memory in most environments once the interaction exceeds 5 steps (see Figures 8 and 9).

## D.1 HALLUCINATING PAST OBSERVATIONS DURING BELIEF UPDATE REASONING

We provide an example ABBEL belief update from Deepseek R1 in Mastermind, demonstrating how LLMs may hallucinate observations from previous steps when reasoning through a belief bottleneck in multi-step environments.

---

**Game State**

**Secret:** 4 5 1 8
**Step:** 5 out of 12
**Goal Reached:** No

---

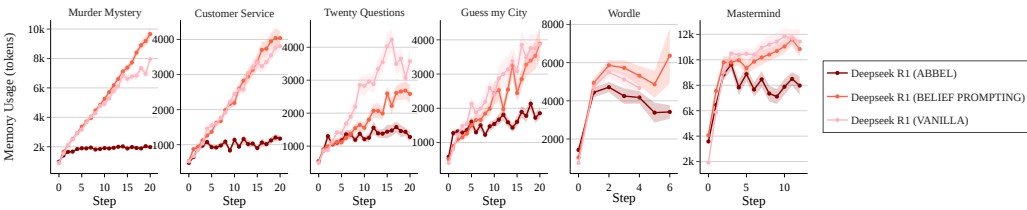

Figure 9: The memory usage at each step, defined as max(input + output tokens for belief updating, input + output tokens for action selection), representing the inference-time memory requirement. After the first few steps, ABBEL uses significantly less memory than the other frameworks.

---

**Prior Belief**

**Exclusions**: 0, 6, and 9 are not in the code.
**Possible Codes**:
- 4 5 [1/2/3] [7/8]
- 4 5 [7/8] [1/2/3]

---

**Action and Observation**

**Action:** 4 5 1 7
**Observation:** 3 exact matches, 0 partial matches

---

**Deepseek R1 (ABBEL) Belief Update Reasoning**

...but previous feedback said that the third guess (4 5 7 8) had a partial match...

---

`4568` was never guessed; this hallucinated previous feedback leads R1 to conclude that 7 was the partial match and must be in position 4, ruling out the true code `4518` from the posterior:

---

**Updated Belief State**

**Exclusions**: 0, 1, 6, 8, 9 are not in the code.
**Possible Codes**:
- 4 5 2 7
- 4 5 3 7

---

### D.2 BELIEF STATE RECONSTRUCTION UNDER VANILLA AND BELIEF PROMPTING

We highlight how frontier models reconstruct previously stated information in the Vanilla and BELIEF PROMPTING settings.

We find that Deepseek R1 reconstructs a posterior belief from the interaction history as the first step of reasoning about the next action:

---

**Deepseek R1 (vanilla) Action Selection Reasoning**

Okay, let's see. So, the history so far has three guesses: `CLOUD`, `PULSE`, and `FAULT`. Let me recap the feedback from each.

- **From `CLOUD`:**
  `C`, `O`, `D` are not in the target. `L` (second letter) is present but wrong position. `U` (fourth letter) is present but wrong position.

---

> - **From PULSE:**
>   P, S, E are not in the target. U (second letter) and L (third letter) are present but wrong positions.
> - **From FAULT:**
>   F, A are not in the target. U (third), L (fourth), and T (fifth) are present but wrong positions.
>
> Putting this together, the letters that are definitely in the target word are L, U, and T. Because in FAULT, even though F and A are excluded, U, L, T are confirmed. Additionally, from CLOUD, L and U are confirmed but not in positions 2 and 4 respectively. From PULSE, U and L are again confirmed but positions 2 and 3 are wrong.
> . . .

We find that Deepseek R1 with belief prompting sometimes fails to leverage the explicit belief state in its context, and wastefully reconstructs a posterior belief from the interaction history:

---

**Deepseek R1 (BELIEF PROMPTING) Action Selection Reasoning**

**Belief State**
First letter is S. Excluded letters: P, O, I, L, E. Correct letters: A in position 3, R in position 4, T in position 5. Target word format: S_ART. Possible letters for position 2: B, C, D, F, G, H, J, K, M, N, Q, U, V, W, X, Y, Z.

**Reasoning**
Okay, so I'm trying to figure out the next guess for Wordle. Let's recap what we know so far. The first guess was SPOIL. The response told us that S is correct in the first position. The letters P, O, I, L, and E (from the fifth letter in SPOIL and the fifth in STARE) are excluded. Then the second guess was STARE, which confirmed that S is still correct in position 1. The letter T is present but not in position 2. Also, A is correct in position 3, R in position 4, and E is excluded. Now the target word structure is S _ A R T. Wait, no, wait. Wait, let me check again.

Wait, in STARE, the fifth letter was E, which is excluded. But the response for STARE said that the fifth letter E is not in the target word. So the target word structure is S _ A R _, since the fifth letter isn't E. Wait, hold on. No, the correct letters from STARE: third letter A (position 3), fourth letter R (position 4). The fifth letter E is excluded, so the fifth position can't be E. But the first letter is S. The second letter is T in STARE, but it's in the target word in a different position. So T exists but isn't in position 2. So the target word has T somewhere else. But the structure we have is S _ A R _, with the fifth letter not E, but from the first guess SPOIL, the fifth letter L is excluded. So fifth position can't be L or E. Wait, but what about the third guess? Wait, the user only provided two guesses: SPOIL and STARE. Then the belief state says the target word is S _ A R T. Wait, how did T get into the fifth position? Because in STARE, the fifth letter E was wrong, but the correct letters in STARE are S (position 1), A (3), R (4). Then T in position 2 is present but in a different spot. So T must be in position 5? Because the only other positions are 2 (which was T in STARE but it's not there), or 5. Because positions 1, 3, 4 are already S, A, R. So T must be in position 5. That makes the target word S _ A R T. So fifth letter is T. So structure is S _ A R T.
. . .

---

Finally, we find that Deepseek R1 under ABBEL reasons directly with the belief state:

---

**Deepseek R1 (ABBEL) Action Selection Reasoning**

**Belief State**
Excluded letters: S, L, T, R, C, D, H, V
Confirmed present letters: A (position 2), E (position 4), N (position 5)
Possible letter positions:
1: [B, F, G, I, J, K, M, O, P, Q, U, W, X, Y, Z]

---

```
2: [A]
3: [B, F, G, I, J, K, M, O, P, Q, U, W, X, Y, Z]
4: [E]
5: [N]
```

**Reasoning** Okay, let's see. I need to figure out the best word to guess for the third attempt in Wordle. The excluded letters are S, L, T, R, C, D. So I can't use any of those. The confirmed letters are A in position 2 and E in position 4. So the word must be _ A _ E _.
My first thought is to maximize the information gain. Since this is the third attempt and there are three more left, I shouldn't be too greedy yet. I need to test possible letters in the unknown positions (1, 3, 5) while using letters that aren't excluded.
Looking at the possible letters for positions 1, 3, and 5: they include B, F, G, H, I, J, K, M, N, O, P, Q, U, V, W, X, Y, Z. I should choose a word that uses as many of these as possible in the unknown slots to check which ones are present.
. . .

## E    BELIEF STATE GRADING EXAMPLES

Early on in training belief states were messy and rarely parse-able:

---
**Prompt**

You will determine the correct combination of characters at [Position 1, Position 2, Position 3] in a 3-character combination lock through iterative reasoning and queries.
All 3 characters are unique.
The set of valid characters are as follows: ['0', '1', '2', '3', '4', '5', '6', '7', '8', '9']
Each action is a query of the form ['char 1', 'char 2', 'char 3'].
Each time you query a combination, you will get feedback from the user about each character: either not in the combination, in the combination but in a different position, or in the combination and in the right position.
You can make up to 12 queries.
Your goal is to find the correct combination in the least number of queries.
Your current belief state: <belief>No prior belief.</belief>
Your last action:
<action>['0', '1', '2']</action>
Environment feedback:
0 is not in Position 1, but is in the lock
1 is not in the lock
2 is not in the lock
Now update your belief state to include all important new information you have gathered.
Do not say anything about future actions. Think step by step and then output your new belief state inside <belief> ... </belief>, e.g., <think>Any thinking</think><belief>your new beliefs</belief>.

---
**Response**

<think>Given the feedback from the query ['0', '1', '2']:
- '0' is not in Position 1 but is in the lock.
- '1' and '2' are not in the lock.
This means '0' must be in either Position 2 or Position 3, and the characters for Position 1, 2, and 3 are now limited to ['0', '1', '2'] but with '1' and '2' not in Position 1. Therefore, '0' is the only character that can be in Position 1, Position 2, or Position 3 excluding the position it is not in.
So, Position 1 cannot be '0', and it must be one of '1' or '2'. Positions 2 and 3 can be either '0', '1', or '2', but '1' and '2' cannot be in Position 1.</think>
<belief>Position 1 can be '1' or '2'.

---

> Position 2 and Position 3 can be '0', '1', or '2', but '1' and '2' cannot simultaneously be in Position 1 and the other positions at the same time.
> '0' must be in either Position 2 or Position 3.</belief>

Near the end of the belief state grading training run, we find belief states become more parse-able by our parsing model grok 4 fast (free) provided by openrouter, and the model is more able to handle complex updates to its posterior.

---

**Prompt**

You will determine the correct combination of characters at [Position 1, Position 2, Position 3] in a 3-character combination lock through iterative reasoning and queries. All 3 characters are unique.
The set of valid characters are as follows: ['0', '1', '2', '3', '4', '5', '6', '7', '8', '9'] Each action is a query of the form ['char 1', 'char 2', 'char 3'].
Each time you query a combination, you will get feedback from the user about each character: either not in the combination, in the combination but in a different position, or in the combination and in the right position.
You can make up to 12 queries.
Your goal is to find the correct combination in the least number of queries.
Your current belief state: <belief>given that '2' cannot be in position 3, it must be in position 1 or position 2. the valid characters are now ['2', '3', '4', '5', '6', '7', '8', '9']. '0' and '1' are not in the combination. the remaining characters need to be tested to determine their correct positions.</belief>
Your last action:
<action>['2', '3', '4']</action>
Environment feedback:
2 is not in Position 1, but is in the lock
3 is not in the lock
4 is not in the lock
Now update your belief state to include all important new information you have gathered.
Do not say anything about future actions. Think step by step and then output your new belief state inside <belief> ... </belief>, e.g., <think>Any thinking</think><belief>your new beliefs</belief>.

---

**Response**

<think>From the feedback, we know:
- '2' is in the combination but not in Position 1, which means '2' must be in Position 2.
- '3' is not in the combination, so we can exclude it from the valid characters.
- '4' is not in the combination, so we can exclude it from the valid characters.
The valid characters are now further reduced to ['2', '5', '6', '7', '8', '9']. Since '2' is fixed in Position 2, the remaining characters need to be tested in both Position 1 and Position 3 to determine their correct positions.</think>
<belief>Given the feedback, '2' is fixed in Position 2. The valid characters are now ['2', '5', '6', '7', '8', '9']. '3' and '4' are not in the combination. The remaining characters need to be tested to determine their correct positions in Position 1 and Position 3.</belief>

---

## F   16 OBJECTIVE QA EXAMPLES

---

**ABBEL**

**Questions**
The principal author of the US Constitution and writer of over 1/3 of the Federalist Papers, which US President, the 4th, was CIC during the War of 1812?; Of what modern country is Sarajevo the captial of?; Pre restraining order(s), who did People magazine name as their first

---

"Sexiest Man Alive", in 1985?;... What 70's group released a statement on Monday that read, in part "We are deeply dismayed by today's announcement from the YMCA that they feel a name change and a rebranding are in order after 166 years"?

**Belief State**

<belief>James Madison was the 4th US President and was CIC during the War of 1812. We need to find the "Sexiest Man Alive" in 1985, the part of the human body the carotid arteries supply, the innovation introduced by the Chillicothe Baking Company, the hand holding the torch on the Statue of Liberty, the sense referred to by the term gustatory, the canal construction started on July 4, 1817, the baseball legend who retired in 1939, the 2007 movie based on the plot summary, the Latin phrase for something designed for a specific problem or task, the planet with Phobos and Deimos as moons, the number of blackbirds in the nursery rhyme, the proverb, the two cities in Charles Dickens' novel, and the 70's group that released a statement about the YMCA.</belief>

MEM1 does not separate the belief from the reasoning, which can lead to longer internal states.

---

**MEM1-Instruct**

**Questions**

Who is James Cecil, 6Th Earl Of Salisbury's paternal grandfather?; ... Where did the director of film The Woman In The Hall study?; What is the place of birth of the composer of film Chakravalam Chuvannappol?; Where did Africanus Fabius Maximus's father die?; Which film has the director who is older than the other, Sign Of The Pagan or Spawn Of The North?

**Internal State**

<think>From the documents, we can infer that Jacques Becker, the director of "Edward And Caroline," was born on 15 September 1906 in France. The place of birth of the composer of "Chakravalam Chuvannappol" is not provided, so we will need to search for more information about the composer of this film. The father of Africanus Fabius Maximus is not provided, so we will need to search for more information about Africanus Fabius Maximus. The director of "The Woman In The Hall" is not provided, so we will need to search for more information about the director of this film. The place of death of Anne Devereux's husband is not provided, so we will need to search for more information about Anne Devereux. The directors of "Friday The 13Th (1916 Film)" and "Beaumarchais (Film)" are not provided, so we will need to search for more information about these directors. The film "Sign Of The Pagan" was released in 1935, and "Spawn Of The North" was released in 1934. Therefore, the director of "Sign Of The Pagan" is older than the other.</think>

Training ABBEL with a length penalty leads to even more concise beliefs that do not enumerate all the unknown information.

---

**ABBEL-Length-Penalty**

**Questions**

What literary style describes the 1968 book that features a 1939 international Harvester school bus?; Is the building located at 200 West Street taller than the one at 888 7th Avenue?; ... Are Waste Land and Shikashika both types of drinks?

**Belief State**

<belief>The 1968 book "The Electric Kool-Aid Acid Test" by Tom Wolfe is written in the New Journalism style. The building at 110 West 7th Street is taller than the one at 888 7th Avenue.</belief>

---

# G  RL DETAILS

## G.1  COMBINATION LOCK ENVIRONMENT DETAILS

*Combination Lock* has the same feedback dynamics as *Wordle* with 3-character codes and guesses, while additionally enforcing that all three characters of the secret code and of every guess must

be unique. Unique secret codes of 3 vocabulary characters were sampled, with a larger disjoint vocabulary and increased horizon at test time (see Table 4).

Table 4: Characteristics of the *Combination Lock* environments.

| Setting | Horizon (H) | Vocabulary | Answer Space Size |
|---------|-------------|------------|-------------------|
| Train | 12 | 012345689 | 720 (3 unique digits) |
| Test | 16 | qawsedrftgyhujik | 3360 (3 unique letters) |

We prompted Qwen2.5-7B-Instruct to first think step by step between `<think>...</think>` tags, and then generate actions or beliefs between `<action></action>` or `<belief>...</belief>` tags. Invalid generations did not count as an environment step, i.e. did not impact regret, but we limited the number of generation calls per game to $H$ (VANILLA) or $2H$ (ABBEL and BELIEF PROMPTING); see Table 5 for details. Each trajectory ends in success once the secret code is guessed, or failure if either the generation limit or environment horizon is exceeded, with reward defined as follows to encourage succeeding with as few guesses as possible:

$$\mathcal{R} = \begin{cases} (H + 1 - \text{environment steps taken})/H & \text{if trajectory successful} \\ -1 & \text{otherwise.} \end{cases} \quad (2)$$

## G.2 COMBINATION LOCK TRAINING DETAILS

See Table 6 for the training settings and hyper parameters used, and Algorithm 2 for the belief grading algorithm.

---

**Algorithm 2** GRPO with Belief Grading

---

**Require:** Environment instructions $p_I$; belief generation prompt $p_b$; belief parsing prompt $p_p$.
**Require:** ABBEL policy model $\pi_\theta$; batch of trajectories $\{\tau_i\}$ rolled out by $\pi_\theta$; belief parser $\Pi$.
   belief_groups $\leftarrow [\,]$
   **for** traj in $\{\tau_i\}$ **do**
      **for** $t$, step in enumerate(traj) **do**
         $(b_t, a_t, o_t, b_{t+1}) \leftarrow$ step
         belief_context $\leftarrow p_I, b_t, a_t, o_t, p_b$
         $b'_{t+1} \sim \pi_\theta(\cdot|\text{belief\_context})$          ▷ Redo belief update generation at this step.
         $r \leftarrow \text{GRADE\_BELIEF}(b_{t+1}, \text{traj}, t)$
         $r' \leftarrow \text{GRADE\_BELIEF}(b'_{t+1}, \text{traj}, t)$
         belief_group $\leftarrow [(\text{belief\_context}, b_{t+1}, r), (\text{belief\_context}, b'_{t+1}, r')]$
         belief_groups.append(belief_group)
         **if** $r = 0$ **then**
            **break**          ▷ Go to next trajectory after the first incorrect belief
         **end if**
      **end for**
   **end for**
   Add belief_groups to the current batch of trajectory groups.
   Update $\pi_\theta$ on all groups with GRPO.
   **function** GRADE_BELIEF($b_{t+1}$, traj, $t$)
      $b^*_{t+1} \leftarrow \text{compute\_posterior}(\text{traj}[:t])$       ▷ Get true posterior from info in previous steps.
      parsed_belief $\sim \Pi(\cdot|p_p, b_t)$       ▷ Parse $b_{t+1}$ into the same format as $b^*_{t+1}$.
      **return** parsed_belief $= b^*_{t+1}$       ▷ Return reward of 1 if $b_{t+1}$ is correct.
   **end function**

---

## G.3 QA TRAINING DETAILS

See Table 7 for the training settings and hyper parameters used.

Table 5: Handling of invalid generations in *Combination Lock*.

| Case | Description | Outcome |
|------|-------------|---------|
| Valid action | The action generation is correctly formatted as `<action>[c1, c2, c3]</action>` with three unique characters. | Both generation and environment steps are incremented, and feedback is presented in a newline separated list. e.g,:
`8 is in Position 1!`
`6 is not in Position 2, but is in the lock`
`9 is not in the lock` |
| Invalid action | Most often errors take the form of `[action>...</action>` or repeated characters. | Generation step is incremented, and the model receives a message stating the action is invalid, reiterating the required format and prompting regeneration. |
| Invalid belief | Not using `<belief></belief>` tags. Errors tend to result from forgotten beginning/ending angle brackets or misspellings of `belief`. | Generation step is incremented, and the model receives a message stating the belief is invalid, reiterating the required format and prompting regeneration. |

Table 6: Settings used in *Combination Lock* experiments. The mini batch at every gradient update step was set to the number of tensors present in the step to prevent off-policy updates, which have been shown to result in unstable training behavior with Qwen models.

| Name | value |
|------|-------|
| Optimization Algorithm | GRPO |
| AdamW learning rate | 1e-7 |
| batch_size | 16 |
| GRPO n rollouts | 2 |
| mini_batch | N/A |
| training_steps | 140 |
| num_epochs (calculated equivalent) | 3.2 |
| Learning rate decay | 0.0 |
| Gradient clipping | 1.0 |

**Belief Length Penalty** To calculate the penalty for a trajectory, we take the length of longest belief state in the trajectory, subtract the mean over all trajectories in the batch, and apply a 0.01 scaling factor. We only apply a penalty to trajectories which do create a valid belief state, so as not to reward generating empty beliefs. In addition, we do not normalize the lengths by the in-batch range, and apply the penalty after advantage normalization, such that as the belief states get shorter the penalty has a smaller impact. We found this was important to avoid over-compressed beliefs significantly harming performance.

We find that the peak token metric isn't very precise, and should instead control for the step at which the agent is at. More steps of information collection require more tokens in the belief state resulting in higher penalties, meaning the model desires to reduce its searches. In QA, the model may opt to depend on its parametric knowledge in place of searches as a strategy to reduce its task reward, which fails to capture our desire, but will minimize this metric.

### G.4 COLBENCH DETAILS

**Example ABBEL-Belief-Grading Belief State at Step 50**

The user expects the function to handle edge cases where revenue and variable costs are equal. Specifically, if revenue and variable costs are both USD100,000, the break-even point should be very high, potentially approaching infinity, and the margin of safety to be close to $100\%$.

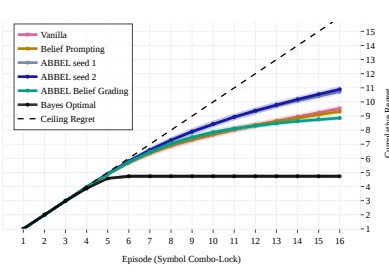

(a) Test set mean cumulative regret.

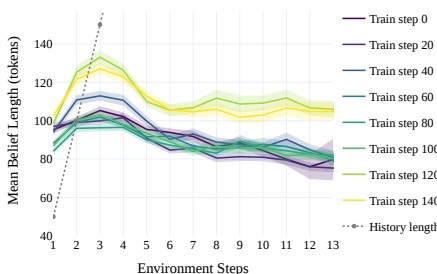

(b) ABBEL .

Figure 10: Test behavior of Qwen2.5-7B-Instruct trained in *Combination Lock*. (a) Final cumulative regret shows that after training, ABBEL still takes more attempts on average to find the secret code than models trained with access to the full history in context (VANILLA and BELIEF PROMPTING). However, when augmented with belief grading, ABBEL outperforms these settings. (b) ABBEL without belief grading learns to generate longer beliefs, but they remain significantly shorter than the interaction history beyond the first two environment step.

Table 7: Settings used in QA experiments.

| Name | value |
|------|-------|
| Optimization Algorithm | GRPO |
| AdamW learning rate | 1e-7 |
| batch_size | 16 |
| GRPO n rollouts | 2 |
| mini_batch | N/A |
| training_steps | 260 |
| num_epochs (calculated equivalent) | 3.2 |
| Learning rate decay | 0.0 |
| Gradient clipping | 1.0 |

They consider a margin of safety of $100\%$ as a reasonable way to represent a break-even or nearly break-even situation. The function should output two numbers as the break-even point and margin of safety, even in edge cases. The function should calculate the margin of safety as $((revenue - (fixed\_costs/(1 - (variable\_costs/revenue))))/revenue) * 100\%$, representing how much sales can drop before incurring a loss. The function signature is: $def\,calculate\_break\_even\_point(revenue, fixed\_costs, variable\_costs)$.

**Example ABBEL (No Belief Grading) Belief State at Step 50**

Target year: 2050, Reduction percentage: $50\%$, Current emissions data: symbolic variables (e.g., $current\_emissions$), Clarification needed: total emissions cut by 2050 or annual reduction rate.

