# OpenReview forum: "ABBEL: LLM Agents Acting Through Belief Bottlenecks Expressed in Language"
_ICLR.cc/2026/Conference — Submitted to ICLR 2026_

### Official Review · Reviewer_dYtH · 2025-10-27

**Soundness:** 2
**Presentation:** 2
**Contribution:** 3
**Rating:** 4
**Confidence:** 4

**Summary:**

The paper proposes regularly asking a language model to express its current belief state in natural language as a way to alleviate degradation caused by context length in long interactions. First, it evaluates frontier models in their ability to summarize context sufficiently into a belief state, identifying some failure modes like propagation of incorrect beliefs, and repeating uninformative actions because they don't reflect as changes in belief states. Then, it uses GRPO (group size 2) train LLMs to better generate and use belief states. Direct training improves LLM use of belief states, but does not match full context training. To further improve belief states, the paper proposes belief grading rewards, which can then outperform full history training on simple multi-turn tasks.

**Strengths:**

1. Training LLMs to produce and use summaries of context to alleviate long-context issues and costs is an interesting and important direction.

2. The analysis on how belief state compaction, while somewhat effective, still has failures modes without training is helpful, and creates a clear logical flow.

3. It is good that the paper considers both outcome reward training, but also a stronger (albeit seemingly very task specific) way to grade the beliefs themselves.

**Weaknesses:**

1. The core idea of summarizing LLM context has a long line of prior work [e.g. 1,2,3] which is not discussed and should be implemented and compared as baselines in this work. The only baseline compared is Mem1, however the methodology and results here are questionable. For example, the baseline is not reproduced in the paper's evaluation setting, and instead numbers are taken directly from the paper. Figure 7(b) is referenced as showing Mem1 states being "much longer", but actually has no Mem1 results! In fact, in 7(c), Mem1 seems to have lower token usage.

2. The "best" method, "belief grading" is not described clearly. It seems like it involves using a much more capable model to perform analysis very specific to the Combination Lock task (simplified version of wordle with just 3 digits). How generalizable is this method? Why was it not used in the multi objective QA environment? This is important because without belief grading, the RL method performs worse than full context RL on the task. Perhaps switching to a more suitable and realistic task which actually tests long context capabilities would be better to demonstrate the benefits of outcome reward RL for generating and using belief states?

[1] LongLLMLingua: Accelerating and Enhancing LLMs in Long Context Scenarios via Prompt Compression
Huiqiang Jiang, Qianhui Wu, Xufang Luo, Dongsheng Li, Chin-Yew Lin, Yuqing Yang, Lili Qiu.

[2] MemGPT: Towards LLMs as Operating Systems
Charles Packer, Sarah Wooders, Kevin Lin, Vivian Fang, Shishir G. Patil, Ion Stoica, Joseph E. Gonzalez

[3] Recursively Summarizing Enables Long-Term Dialogue Memory in Large Language Models
Qingyue Wang, Yanhe Fu, Yanan Cao, Shuai Wang, Zhiliang Tian, Liang Ding

**Questions:**

Main question: How exactly is belief grading implemented, and how generalizable is this methodology for other, realistic tasks? Which component of it helps the most, the supervision signal from a much stronger model, or something else? How does it compare to distillation from the stronger model?

Suggestion: Figure 7 is hard to read due to the extremely thin bars. Consider making the X axis (number of objectives) log-scale.

Suggestion: The paper keeps switching environments across sections by just citing the works that proposed them. It would be helpful for readers if the paper is self-contained, and each section begins with a clearer description of the environment being used.

Nitpick: "multi-objective QA" is called "realistic" in the discussion, when its a highly synthetic task obtained from randomly combining QA from existing retrieval datasets. I would refrain from calling this environment realistic throughout the paper.

Question: Why were the same environments not used throughout? For example why not train/test for the environments used in Figure 4, which are much more interesting than Combination Lock?

---

> ### Author Response · Authors · 2025-11-25
> **Response part 1**
>
> We thank reviewer dYtH for their thoughtful comments and helpful suggestions. We are glad that they agree we tackle an interesting and important problem, and appreciate the clear logical flow of our paper. We address their weaknesses and questions below:
>
> >summarizing LLM context has a long line of prior work [e.g. LongLLMLingua,MemGPT,Recursive Summarization] which is not discussed and should be implemented and compared as baselines
>
> Thank you for bringing these works to our attention, we have now added all of them to our related work section. However, MemGPT generates an external memory store, which is an orthogonal approach that faces a very different set of constraints and can be combined with ABBEL for greater gains. While LongLLMLingua and Recursive Summarization also summarize contexts without relying on external memory, they focus on the single-step setting where the context is static, e.g. a long document or collection of dialogs. We study the more general multi-step setting where the agent must maintain a summary while actively exploring. In our setting, the agent must reason over its beliefs to select actions that gather missing information needed for the task; whereas an agent in a single-step environment need not reason about this because all information is already given. Please let us know if you know of any other missing prior work.
>
> >[Mem1] is not reproduced in the paper's evaluation setting, and instead numbers are taken directly from the paper. Figure 7(b) is referenced as showing Mem1 states being "much longer", but actually has no Mem1 results!
>
> We would like to clarify that we did reproduce MEM1: we re-implemented MEM1 in our own codebase and also trained it with Qwen-7B-Instruct (the same model we used for ABBEL), reporting it as “MEM1-Instruct” in orange in the figure (now Figure 8 in the revised manuscript). When we reran MEM1’s code directly, we found that it actually underperformed compared to the results reported in their paper, so we had to take the numbers directly from their paper (which we include as just “MEM1”, in red in Fig. 8) but we note that we evaluate our models in the exact same setting as MEM1. This also motivated our reimplementation of MEM1, to give it the fairest chance in comparison to ABBEL: our reimplementation actually performs a bit better than the original MEM1 paper’s reported results.
>
> >In fact, Mem1 seems to have lower token usage.
>
> In our original submission, in only the 8-objective setting MEM1 did report slightly lower token usage than even ABBEL-Length-Penalty. However, we have since realized we were inappropriately normalizing the length penalty, which prevented its magnitude from naturally decreasing relative to outcome rewards as the beliefs get shorter, so we retrained without normalization and found the performance increased to close to the original ABBEL model. Though the beliefs are somewhat longer than the original length penalty model, they are still significantly shorter than all other models, and they contain more useful information which enabled better reasoning and actually led to lower peak token usage overall, so now ABBEL-Length-Penalty beats MEM1 by a significant margin for all settings with more than 2 objectives. Please see the revised manuscript for the new results.

---

> ### Author Response · Authors · 2025-11-25
> **Response part 2**
>
> >switching to a more suitable and realistic task which actually tests long context capabilities would be better
>
> We believe that the QA task does test long context capabilities, as Qwen-14B-Instruct was unable to handle the 16 objective setting despite the full interaction history fitting in its context. Though we agree that it is not a very realistic task, and we now additionally evaluate ABBEL in ColBench (please see our overall response).
>
> >Figure 7 is hard to read due to the extremely thin bars ;It would be helpful for readers if the paper is self-contained, and each section begins with a clearer description of the environment being used.
>
> Thank you for these suggestions for improving the clarity of our paper, we have remade that figure (now Figure 5 in the revised manuscript) and added clearer descriptions of the environments at the beginning of each section.
>
> >"multi-objective QA" is called "realistic"
>
> We agree with the reviewer’s point, and have updated the manuscript accordingly.
>
> >why not train/test for the environments used in Figure 4, which are much more interesting than Combination Lock?
>
> Combination lock is a simplified version of Mastermind and Wordle from Figure 4 (now Figure 2 in the revised manuscript), and still involves multi-step information gathering and sophisticated reasoning where observations can have non-local implications requiring multiple reasoning steps (e.g., feedback that the “4” guessed at position 1 is not in the combination makes it possible to conclude that position 2 must be “6”, which then excludes “6” from the posterior belief at position 3). Thus, it is a useful testbed for our approach before moving to more realistic environments. All of the other environments in that figure use gpt-o4-mini to simulate humans, which would exceed our budget to query for sufficient steps to do RL.
>
> **Belief grading**:
>
> >"belief grading" is not described clearly. How exactly is it implemented?
>
> Thank you for pointing this out. We have added thorough explanations of belief grading to the revised manuscript in section 5.1.2, including a diagram and pseudocode algorithm; we believe it is now a much clearer description. In short, we treat belief generation as a separate task, creating additional "trajectories" consisting of single belief update generations to which we assign the grading rewards. To do this, we collect belief states generated during the environment's multi-step roll out, and for each belief we prompt ABBEL again to generate another belief from the same context to create a size-2 group for GRPO. We then use a grading function to assign rewards to each belief, which provides a learning signal whenever the two beliefs in a group receive different grades. The policy gradient step for belief grades and outcome-based rewards is applied concurrently.
>
> Different grading functions may be used for different domains. In Combination Lock it was defined as follows: for each belief, we first used Grok-4-Fast-Free to parse it into a list of possible numbers at each position, which we compared to the ground truth posterior (computable from the previous observations in the trajectory), generating reward 1 when they were identical and 0 otherwise. We stop grading each trajectory after the first step with an incorrect belief, to avoid penalizing beliefs that were only incorrect due to propagating errors from earlier steps.
>
> >Which component of it helps the most, the supervision signal from a much stronger model, or something else?
>
> For our combination lock belief grader, the stronger language model was used only for parsing the beliefs into a simple format that can be easily compared with the ground truth beliefs; we believe the most helpful component was the supervision from the ground truth/Bayes-optimal posterior distribution, which we calculated algorithmically.
>
> >How generalizable is belief grading?
>
> The belief grading framework for augmenting the outcome-based reward during RL training through belief bottlenecks is highly general, though the choice of function used to grade each belief may be task-specific. The grading function we used in Combination Lock was highly specific to the game, but there are also fully general belief grading functions. Please see our overall response for a domain-general heuristic we proposed that does not rely on ground truth posteriors or LLM parsers, that we found helped ABBEL learn in a setting where such posteriors were unavailable.
>
> >Why was belief grading not used in the multi objective QA environment?
>
> We did not use it in multi-objective QA because the belief state update for QA does not involve as much complex reasoning as in Combination Lock, while QA’s lengthier observations made it a more ideal environment for studying belief length penalties for improving memory efficiency.

---

> > ### Comment · Reviewer_dYtH · 2025-11-25
> >
> > Thanks for the detailed response and new experiment on ColBench. I am a bit confused by the result reported in Table 2, as ABBEL seems to be significantly worse than Vanilla, though I do understand it reduces the token usage.
> >
> > Overall, my main concern about the lack of baselines in this work has not been addressed. For e.g., even if past related work was on single-turn tasks, the methods can still apply to the multi-turn setting, and are thus relevant baselines. In comparisons with Vanilla RL training, ABBEL does worse but uses less tokens. Another important missing baseline here is Vanilla RL with length penalties, which has been extensively explored and is also a simpler solution than ABBEL.

---

> ### Author Response · Authors · 2025-11-26
>
> We thank reviewer dYtH for taking the time to consider our response, and greatly appreciate their very prompt reply.
>
> Apologies for the confusion about ABBEL’s performance on ColBench- we said in our original overall response that “ABBEL is still close to the performance of the full context model” but to be precise, ABBEL reaches 88% of the performance of the full context model while using 49% the memory. We have clarified the wording of our overall response.
>
> >even if past related work was on single-turn tasks, the methods can still apply to the multi-turn setting, and are thus relevant baselines
>
> We would like to further clarify how these methods for single-turn tasks cannot apply to the multi-turn setting. These works do not specify how to update summaries with new actions and observations after each step, and adapting them to the multi-turn setting would require extensive modification to the point that they would be more like follow-up works than reimplementations.
>
> E.g., for Recursive Summarization to generate a single action (response), it summarizes a list of dialogue sessions, adding one session at a time to the memory. Our settings generate an action after every observation- how would a single observation (e.g., a 2-sentence user message in ColBench) be divided into a list of sessions to be recursively summarized? If we modify it to treat each action and observation as an entire “session” to be added to memory from the previous step's memory, this actually corresponds to ABBEL without RL.
>
> LongLLMLingua compresses a set of text documents to generate an action, where their key contributions include coarse-to-fine summarization and document reordering. This also does not apply to settings with short observations between actions like ColBench, and they provide no method for compressing actions and previously generated compressed contexts along with the new documents retrieved at each step- e.g., in Multi-Objective QA how can they order all the documents by relevance if the compressed context cannot be separated back out into all the documents that originally went into it? Should the action (query) also be treated as a “document”? We also note that the compressed contexts that LongLLMLingua generates are uninterpretable, while we study the problem of generating human-understandable language summaries.
>
> In general, methods for the single step setting involve complex processing of a long context before generating one action, whereas in multi step settings the initial context may be short and the length may only come from adding up many steps of interaction. We hope this clarifies why these baselines do not apply to our setting, and also highlights why work like ours is important as it investigates an understudied setting.
>
> >In comparisons with Vanilla RL training, ABBEL does worse but uses less tokens.
>
> We do not believe this invalidates our findings- it is natural that such significant memory reduction would come at a performance cost, and it is notable how low this cost is for such a large reduction. We also clarify that our aim is not to outperform Vanilla, since Vanilla does not address the problem of context management for very long horizon settings. Over the shorter horizons we evaluate in, Vanilla simply serves as a best-case point of comparison for us to study the behavior of LLMs acting through context summaries. To the best of our knowledge, we are the first to show that RL with outcome rewards alone can be insufficient for an LLM to learn to generate sufficient context summaries in multistep settings (i.e., to perform as well as when trained with access to the full context). This motivates us to propose belief grading, which even outperformed Vanilla in Combination Lock, and sets the stage for future research on this important problem.
>
> >Another important missing baseline here is Vanilla RL with length penalties, which has been extensively explored and is also a simpler solution than ABBEL.
>
> We appreciate the suggestion of an additional baseline. However, we do not believe that Vanilla RL with length penalties is a solution to context management for long horizon interaction. No matter how much we penalize and reduce the length of Vanilla’s generated text (the reasoning and actions), the context length will still grow linearly with each environment observation. E.g., for QA the context will grow by at least 300 words from the retrieved documents at each step, i.e., about 2000 tokens after only 5 steps (twice as long as ABBEL at 16 objectives), while ABBEL’s context grows sublinearly with the number of steps.

---

> > ### Comment · Reviewer_dYtH · 2025-11-27
> >
> > Thanks for your response. Based on the new results provided in the earlier message, and your new clarifications, I have increased the score from 4 to 6.
> >
> > I think overall this work presents a strong idea: RL for belief summarization in long multi-turn interactions. However, the exact method is not fully baked and clear, requiring different belief grading mechanisms for different tasks. The tested tasks also seem somewhat preliminary, consisting of either toy settings or synthetic data multi-turn interactions. The empirical improvements are not that strong, not convincingly better than baseline RL. I think this is a problem with the environments used, but I do sympathize with the authors that open-source environments for long multi turn interactions are lacking.
> >
> > But for the importance of the idea alone, and the proof of concept results, I think the paper should be accepted.

---

> > > ### Author Response · Authors · 2025-12-02
> > >
> > > We are glad that our new clarifications and results were helpful for the reviewer to appreciate the importance and strength of our idea, such that they now agree our paper should be accepted.
> > >
> > > >the exact method is not fully baked and clear, requiring different belief grading mechanisms for different tasks.
> > >
> > > We would firstly like to note that the reconstruction-based belief grader we used in ColBench is fully domain-general and may be used for any task. Secondly, it is a benefit, rather than a flaw, that our belief grading framework provides the flexibility to use different graders for different tasks. Where there is domain knowledge available, it can easily be leveraged by swapping in a domain-specific grader to provide stronger supervision, allowing for even greater performance and training efficiency gains as we saw in Combination Lock.
> > >
> > > >I do sympathize with the authors that open-source environments for long multi turn interactions are lacking.
> > >
> > > We hope that our work can help stimulate the development of more environments with which to study this important problem.
> > >
> > > We thank reviewer dYtH for a highly productive and engaged discussion!

---

### Official Review · Reviewer_y4kX · 2025-10-31

**Soundness:** 2
**Presentation:** 2
**Contribution:** 2
**Rating:** 4
**Confidence:** 4

**Summary:**

- This work proposes using prompt-based belief states to summarize long-context information from LLM-user interactions, and selecting the next actions based only on these belief states rather than conditioning on the full context.
- Shows that without training, the proposed belief-bottlenecked approach ("ABEEL") under-performs compared to full-context or belief-prompting baselines (where both the context and the belief states are available).
- Shows empirically with Qwen 7B-Instruct that RL post-training has some potential to improve ABEEL compared to ABEEL without training on combination lock and the multi-objective QA task.

**Strengths:**

- Improving the performance of LLMs in multi-turn interactions is an interesting problem, but this reviewer is not fully convinced of the novelty or significance of this work due to limited empirical demonstrations (see Weaknesses).
- Clarity: The writing is clear, and Figure 1 clearly illustrates the difference between ABEEL and the existing approaches (vanilla and belief prompting).

**Weaknesses:**

1: Questions about the effectiveness of belief-bottlenecked policies:
- Figure 2 shows that belief-bottlenecked models perform significantly worse than full interaction-based models or models that incorporate both belief and past history. Given the efforts and advances in increasing context lengths for newer models, it is unclear what advantages belief-bottlenecked models offer that long-context models cannot handle. The performance improvement achieved through post-training (when comparing among belief-bottlenecked policies) is insufficient to establish their effectiveness in multi-turn settings, especially if they are still outperformed by existing long-context models.
- The results would be more compelling if there were some domains that long-context models cannot handle but belief-bottlenecked models can.

2: Questions about generalizability across model types and sizes:
- For the combination lock and multi-objective QA tasks, only one model (Qwen-7B-Instruct) is trained to demonstrate ABEEL and ABEEL-Length Penalty.
- The results would be more convincing if the same experiments were repeated with different models and model sizes to show that the effectiveness of post-training with the ABEEL framework generalizes beyond a single model class or type.

3: Insufficient empirical evidence:
- It is unclear whether the performance differences of ABEEL-Length Penalty compared to the other models are statistically significant. In Figure 7, it is also unclear whether the models were trained with more than one seed.
- For Table 1: since Qwen-7B-instruct model is trained (ABEEL, and ABEEL-Length Penalty), it would be useful to also compare with zero-shot Qwen-7B-instruct  in addition to zero-shot Qwen-14B-instruct.
- The advantages of using ABEEL compared to vanilla and belief prompting are not clearly demonstrated in the experiments (Figure 5a and 5b). In particular, ABEEL (seed 2) performs worse than belief prompting and vanilla prompting, raising questions about the strength of the proposed approach compared to the baselines (see Point 1 raised above).

While this paper addresses an interesting problem of effectively handling long contexts in multi-turn interaction settings with LLMs, it could be significantly improved by addressing the concerns regarding the novelty and significance of the empirical results.

**Questions:**

Questions about multiple training seeds and using models other than Qwen 7B-Instruct are raised in Weaknesses to strengthen claims about empirical performance of ABEEL.

Additionally, there are other multi-turn domains that the authors could consider, where vanilla models have been reported to perform poorly.
- For example, Laban et al., 2025 evaluates models on a suite of tasks such as coding, math, and summarization, where task-relevant information is gradually revealed through multi-turn interactions, so it is crucial to remember and maintain beliefs about key problem details.
- In Zhao et al., 2025, "Long-context retrieval" section may be of particular interest to the authors. This task focuses on accurately retrieving and summarizing relevant information about user preferences in long-context conversations.

For both of these benchmark tasks, one could imagine belief states being beneficial and leading to improved performance compared to naïve long-context models.

- Laban et al., 2025. "LLMs get lost in multi-turn conversation". https://arxiv.org/pdf/2505.06120.
- Zhao et al., 2025. "Do LLMs recognize your preferences? Evaluating personalized preference following in LLMs." https://arxiv.org/pdf/2502.09597

---

> ### Author Response · Authors · 2025-11-25
>
> We thank reviewer y4kX for their thoughtful review and valuable suggestions. We’re glad that they agree we tackle an interesting problem, and that they find our writing to be clear. We respond to the weaknesses and address their questions below:
>
> >it is unclear what advantages belief-bottlenecked models offer that long context models cannot handle
>
> Even long context models will always have some finite context limit beyond which context management methods will be absolutely necessary. Furthermore, recent work [1] finds that long-context models struggle to utilize additional context effectively as the context grows (though this effect may not have shown up in the environments in Figure 2, as the contexts weren’t as long as in real world settings). We would also like to highlight our other findings about belief bottleneck advantages in Section 4, including improved reasoning efficiency and the interpretability of the belief state, making it clearer what information an agent used to make a decision.
>
> >The performance improvement achieved through post-training is insufficient to establish [belief bottleneck] effectiveness in multi-turn settings, especially if they are still outperformed by existing long-context models…
>
> We would like to clarify that ABBEL with belief grading performs significantly better than the long-context baselines trained with access to the full context in Combination Lock (Figure 4a in the revised manuscript).
>
> >The results would be more convincing if the same experiments were repeated with different models and model sizes
>
> Thank you for the helpful suggestion, we are planning to run these experiments and will update as soon as we have results.
>
> >It is unclear whether the performance differences of ABEEL-Length Penalty compared to the other models are statistically significant.
>
> We realized we were inappropriately normalizing the length penalty which prevented its magnitude from naturally decreasing relative to the outcome reward as the beliefs get shorter, so we retrained without normalization and found the performance increased to close to the original ABBEL model (e.g., 3.43+/-0.05 exact match score for 16 objectives), which is significantly higher than MEM1 (2.50+/-0.06). Though the beliefs are somewhat longer than the original length penalty run, they contain more useful information that enabled better reasoning and led to even lower overall peak token usage than before. We have now also run our evaluation on the full test set (in the original submission we had only run it on a random subset due to compute constraints) which reduced the variance of all performance estimates. Please see the revised manuscript for the new results.
>
> >In Figure 7, it is also unclear whether the models were trained with more than one seed.
>
> Thank you for pointing out the lack of clarity - so far each model was trained with one seed but we are in the process of training more seeds and will update as soon as we have results.
>
> >since Qwen-7B-instruct model is trained (ABEEL, and ABEEL-Length Penalty), it would be useful to also compare with zero-shot Qwen-7B-instruct
>
> Thank you for the suggestion, we are planning to run this evaluation and will update as soon as we have results.
>
> >there are other multi-turn domains that the authors could consider,
>
> We thank the reviewer for bringing these other domains to our attention. We agree that it would be helpful to evaluate ABBEL in an additional, more complex multi-turn domain. We have new results in ColBench (please see our overall response) which is quite similar to Laban’s benchmark, involving an initial underspecified prompt requiring multiple questions to uncover the necessary details to complete the task. If we have time, we will also evaluate on Laban’s benchmark.
>
> [1] Nelson F Liu, Kevin Lin, John Hewitt, Ashwin Paranjape, Michele Bevilacqua, Fabio Petroni, and Percy Liang. Lost in the middle: How language models use long contexts.

---

> > ### Author Response · Authors · 2025-11-26
> > **update**
> >
> > >since Qwen-7B-instruct model is trained (ABEEL, and ABEEL-Length Penalty), it would be useful to also compare with zero-shot Qwen-7B-instruct in addition to zero-shot Qwen-14B-instruct.
> >
> > We have finished evaluating the zero-shot Qwen-7B-instruct model, and added it to the manuscript. We find that, similarly to zero-shot Qwen-14B-instruct, it performs very poorly (0.3-0.4 Exact Match count across all numbers of objectives) while still using much more memory than ABBEL (1.5-2x the Peak Tokens). Please see section 5.2.2 in the revised manuscript for more details (zero-shot Qwen-7B-instruct is labeled as Vanilla Zero-Shot). Thank you again for the suggestion!

---

### Official Review · Reviewer_eHNU · 2025-11-01

**Soundness:** 2
**Presentation:** 3
**Contribution:** 2
**Rating:** 4
**Confidence:** 3

**Summary:**

ABBEL replaces growing histories with a compact natural-language belief and acts only on that, giving near-constant memory and shorter reasoning. Zero-shot lags full history, but RL, especially belief grading, recovers or beats it and enables a memory–accuracy trade-off. In multi-objective QA it outperforms MEM1 with a shorter internal state; remaining issues are belief-update errors, hallucinated past steps, and repeated actions when beliefs stall.

**Strengths:**

* Clear, interpretable bottleneck: separating stored “belief” from transient reasoning is simple,
  model-agnostic, and yields near-constant memory across steps while often reducing tokens and
  action-side reasoning.
* Solid empirical sweep and diagnostics: six environments, ablations (vanilla / belief-prompting /
  ABBEL), and candid analysis of failure modes (propagated belief errors, hallucinated past steps).
* RL contributions are practical: outcome-based RL recovers most performance; belief-grading
  reduces regret and allows explicit length–performance trade-offs; competitive/better than MEM1
  in long-horizon QA with a more compact internal state.

**Weaknesses:**

* Novelty/positioning: very close to prior “learned memory” agents (MEM1/VeRL/rLLM); the
  belief–reasoning split reads as incremental rather than fundamentally new. Missing/under-cited
  contemporaries (e.g., MemAgent) weaken SOTA claims.
* Baselines/fairness: QA compares ABBEL-RL (7B) to an untrained 14B full-history model; no
  apples-to-apples 7B full-history RL baseline reported. Combination-Lock gains hinge on a toy
  setting and ground-truth belief grading; generalization to realistic tasks is unclear.
* Practicality: two calls per step (belief + action) and RL/aux-task training cost; belief-quality
  heuristics for real domains are unspecified; observed loops where the agent repeats uninformative
  actions when beliefs don’t update.

**Questions:**

* Did you train and evaluate a 7B full-history (vanilla) RL baseline in QA and Combo-Lock? If so,
  please report; if not, this is needed to isolate bottleneck benefits from RL effects.
* How will you grade beliefs without ground-truth posteriors in realistic settings (retrieval QA, SWE)?
  Would a constrained/structured belief schema (e.g., JSON slots) reduce hallucinations and make
  grading feasible?

---

> ### Author Response · Authors · 2025-11-25
> **response part 1**
>
> We thank reviewer eHNU for their insightful review and helpful suggestions. We respond to some of the weaknesses and address their questions below:
>
> >very close to prior “learned memory” agents (MEM1/VeRL/rLLM); the belief–reasoning split reads as incremental rather than fundamentally new
>
> We would like to clarify that VeRL-agent and rLLM are merely open source training libraries that provide flexibility for users to specify their own custom memory modules, but do not implement learned memory agents themselves. We would additionally like to bring to the reviewer’s attention that MEM1 is also currently under review at ICLR.
>
> Maintaining memory through updating a belief state is well-motivated from fundamental agent literature on recursive state estimation [1, 2], carrying forward precisely the relevant information for decision making. In contrast, at each step MEM1 sees the previous step’s reasoning trace to generate the next action, but in general much of the contents of the reasoning for the prior step is not helpful for selecting the next step’s action, only increasing the memory burden and distracting from the relevant information.
>  The isolated belief state additionally opens up many possibilities for steering the models that are not available to MEM1, such as the belief grading and belief penalties which are also novel contributions of this work. There is also a difference in how loss is computed and propagated in MEM1’s model architecture which causes information leakage during RL training, which we believe may explain why their reported performance is so low despite training with 5x more datapoints and 2.5x more rollouts than us (we fixed this in our reimplementation of MEM1 (MEM1 Instruct) to give it the fairest chance in comparison to ABBEL.) We are happy to expand on this if the reviewer is interested.
>
> >Missing/under-cited contemporaries (e.g., MemAgent) weaken SOTA claims.
>
> Thank you for bringing this to our attention. We have now cited MemAgent in our related work section, along with other recent work mentioned by reviewers, e.g., [3,4]. However, while they also manage long contexts without relying on external memory, they focus on the single-step setting where the context is static, with no method to incorporate new actions and observations into the previous step's summary, and would require such significant modification that they would be more like follow-up works or new methods entirely. E.g., the key idea in MemAgent is to divide given documents into 5000-token chunks and iteratively process them into memory before acting. This clearly does not apply when the action and observation are short, e.g. a 1-sentence agent question and 2-sentence human reply, and the context grows long only from adding up many steps of interaction. In the multi-step setting we study, the agent must also reason over its beliefs to select actions that gather missing information needed for the task; whereas an agent in a single-step environment need not reason about this because all information is already given. We hope this clarifies why these contemporaries do not apply to our setting, and also highlights why work like ours is important as it investigates an understudied setting.
>
> >Baselines/fairness: QA compares ABBEL-RL (7B) to an untrained 14B full-history model; no apples-to-apples 7B full-history RL baseline reported.
>
> Thank you for the suggestion, we have now run a full-history RL baseline in QA and find that it performs only slightly better than ABBEL, with no advantage at 16 objectives, despite using 9.5x more memory. This demonstrates that ABBEL gives huge memory efficiency gains. The results have been added to the revised manuscript.
>
> >generalization to realistic tasks is unclear.
>
> This is a good point, we agree that evaluating ABBEL in less synthetic environments is important to demonstrate the practicality of our framework. We have added results in ColBench- please see the overall response for further details.
>
> >belief-quality heuristics for real domains are unspecified;
>
> We agree that it would be helpful to specify some examples of belief quality heuristics that could be used where the ground truth posterior is unavailable. Please see our overall response for a general purpose heuristic we proposed and found helped ABBEL learn in ColBench.
>
> [1] Kalman, Rudolph Emil. "A new approach to linear filtering and prediction problems." (1960): 35-45.
>
> [2] Bergman, Niclas. Recursive Bayesian estimation: Navigation and tracking applications. Diss. Linköping University, 1999.
>
> [3] LongLLMLingua: Accelerating and Enhancing LLMs in Long Context Scenarios via Prompt Compression Huiqiang Jiang, Qianhui Wu, Xufang Luo, Dongsheng Li, Chin-Yew Lin, Yuqing Yang, Lili Qiu.
>
> [4] Recursively Summarizing Enables Long-Term Dialogue Memory in Large Language Models Qingyue Wang, Yanhe Fu, Yanan Cao, Shuai Wang, Zhiliang Tian, Liang Ding

---

> > ### Author Response · Authors · 2025-11-25
> > **response part 2**
> >
> > >Practicality: two calls per step (belief + action)
> >
> > It would be straightforward to modify ABBEL such that multiple action steps are taken and added to the context between belief updates, and then update the belief on all actions and observations since the last update, amortizing the cost of an additional belief generation call across multiple steps. We use the simpler setting of generating a belief and action at every step as our testbed, in order to study iterated belief updating within the limitations of our compute budget.
> >
> > >Practicality: observed loops where the agent repeats uninformative actions when beliefs don’t update.
> >
> > This is a general problem for memory and action systems, which is not specific to ABBEL. Though this issue should in principle be addressable through RL (i.e., the agent should learn to include failed actions in its belief state) it would also be simple to fix it by keeping the last action in the context, or by taking more than one action step between belief updates.
> >
> > **Questions:**
> >
> > >Did you train and evaluate a 7B full-history (vanilla) RL baseline in QA and Combo-Lock?
> >
> > We did already train and evaluate a 7B full-history RL baseline in Combo-Lock (Vanilla, see Figure 4a). We had not yet in QA at the time of the original submission, but we have now- please see the Baselines/fairness weakness addressed above for QA.
> >
> > >Would a constrained/structured belief schema (e.g., JSON slots) reduce hallucinations and make grading feasible?
> >
> > We thank the reviewer for the interesting suggestion of a structured belief schema, we believe this would likely also be effective and leave it for future work.

---

### Official Review · Reviewer_icj6 · 2025-11-01

**Soundness:** 3
**Presentation:** 2
**Contribution:** 2
**Rating:** 4
**Confidence:** 3

**Summary:**

The paper introduces ABBEL, a framework that replaces growing multi-turn interaction histories with a compact, natural-language belief state. The authors identify frontier models have inferior performance because of error in belief updating. Thus, finetuning LLMs with belief grading improves LLMs in a multi-objective QA environment and shows promising results.

**Strengths:**

1. Usage of belief state to compress the history trajectory to assist LLM for effective actions sampling, making the internal state compact and inspectable.
2. RL with belief grading improves small language model in multi-objective QA tasks. Use a model to parse belief into characters and then comparing with ground truth posterior is a simple yet effective solution.

**Weaknesses:**

1. RL with belief grading heavily relies on ground truth posterior. It’s unclear how robust grading signals will be in complex, non-synthetic settings where ground-truth posteriors aren’t computable. (as mentioned in the "Limitations" sections)
2. Only one benchmark in the main text. I would expect more benchmarks on results of RL + belief grading, like WebShop [1] as the Mem1 [3] authors did, and maybe also ALFWorld [2], a text based environment for agents to reason and interact with.
3. It seems to me that a lot of context is spent on evaluating frontier models with belief bottlenecks. I do believe that this is an important finding for motivation, but I believe more explanations and results can be put in the appendix, where the main text can focus on RL with belief grading.

[1] Yao, Shunyu, et al. "Webshop: Towards scalable real-world web interaction with grounded language agents." Advances in Neural Information Processing Systems 35 (2022): 20744-20757.

[2] Shridhar, Mohit, et al. "Alfworld: Aligning text and embodied environments for interactive learning." arXiv preprint arXiv:2010.03768 (2020).

[3] Zhou, Zijian, et al. "MEM1: Learning to Synergize Memory and Reasoning for Efficient Long-Horizon Agents." arXiv preprint arXiv:2506.15841 (2025).

**Questions:**

1. Would you providing some examples/ideas of applying heuristic function to model ground truth posterior when computing ground truth posterior is not practical?
2. Is the core different between ABBEL and MEM1[1] only about separating belief state update into multiple steps accordingly?

[1]Zhou, Zijian, et al. "MEM1: Learning to Synergize Memory and Reasoning for Efficient Long-Horizon Agents." arXiv preprint arXiv:2506.15841 (2025).

---

> ### Author Response · Authors · 2025-11-25
>
> We thank reviewer icj6 for their thoughtful comments. We are glad that they find our approach to be simple and effective and agree that the results are promising. We respond to the weaknesses and address their questions below:
>
> >Only one benchmark in the main text
>
> We agree that an additional benchmark would help illustrate the general applicability of ABBEL. We have added results in ColBench, a more complex multi-step collaborative task- please see the overall response for further details.
>
> >more [frontier model eval] explanations and results can be put in the appendix,
>
> We appreciate the suggestion, we have condensed and moved some of that section to the appendix to make more space for the RL section, and we fleshed out more explanations, e.g., of belief grading, in the RL section.
>
> >Would you provide some examples/ideas of applying heuristic function to model ground truth posterior when computing ground truth posterior is not practical?
>
> Please see our overall response for one heuristic we applied in ColBench, where the ground truth posterior is not available.
>
> >Is the core difference between ABBEL and MEM1 only about separating belief state update into multiple steps?
>
> As mentioned, ABBEL updates the belief in one step, and then reasons about the action in the next step given the belief state, whereas MEM1 directly reasons about the action given its memory state. However, the most important difference from our approach is that MEM1 uses the action reasoning trace as its memory, i.e., at each action step MEM1 sees the previous step’s reasoning trace and action and immediately reasons to generate the next action. Much of the prior step’s reasoning trace is not relevant for choosing the next step’s action as it contains no additional information about the task, and conditioning on it during action selection only increases the memory burden and distracts from the relevant information. In contrast, ABBEL generates a cleanly separated explicit belief state, so the agent sees only the previous step’s belief state when generating an action. Maintaining memory through updating a belief state is well-motivated from fundamental agent literature on recursive state estimation [1, 2], carrying forward precisely the relevant information for decision making. The explicit belief state also makes methods like belief grading and belief length penalties possible, which can’t be applied to MEM1. We additionally note that MEM1 is also currently under review at ICLR.
>
> There is also a difference in how loss is computed and propagated in MEM1’s model architecture which allows information leakage during RL training, which we believe negatively impacts its performance (we fixed this in our reimplementation of MEM1 (MEM1-Instruct) to give it the fairest chance in comparison to ABBEL.) We are happy to expand on this if the reviewer is interested.
>
> [1] Kalman, Rudolph Emil. "A new approach to linear filtering and prediction problems." (1960): 35-45.
>
> [2] Bergman, Niclas. Recursive Bayesian estimation: Navigation and tracking applications. Diss. Linköping University, 1999.

---

### Author Response · Authors · 2025-11-25
**Overall Response**

We thank all reviewers for their thoughtful comments and helpful suggestions. We are glad that they agree we tackle an interesting (y4kX, dYtH) and important (dYtH) problem, judge our approach to be practical (eHNU) and simple yet effective (icj6), and praise our clear presentation (y4kX, dYtH).

**More Complex Environment, Belief Grading without Ground Truth Posterior**

A common question among the reviewers was whether ABBEL also works in less synthetic settings where the ground truth posterior is unavailable for belief grading. To answer this, we train in ColBench [1], where the agent must help human users write code from an underspecified initial request through asking questions over multiple steps. Here, we propose a fully domain-general belief grading heuristic that does not depend on having access to a ground truth posterior. Newly generated beliefs are graded with respect to how useful they are when used to reconstruct the most recent observation, given the previous belief and action. We compute this directly as the log probability under the agent model of the previous observation given the previous belief, action, and new belief. This results in assigning higher grades (rewards) for beliefs that integrate more information from the most recent observation during each belief update. We find that this helps ABBEL learn faster to generate more informative belief states and perform better in the first 50 training steps than ABBEL without grading, performing on-par with the full context model while using 44% as much memory. After 100 steps, ABBEL is still close to the performance of the full context model (only a 12% reduction in test pass rate) while using 49% of the memory. Please see Section 5.2.3 of the revised manuscript for further details. We hope this additional domain helps demonstrate the efficacy and generalizability of ABBEL and belief grading.

We also highlight that although Combination Lock and QA environments are highly synthetic, they each test ABBEL’s ability to handle specific difficult challenges: Combination Lock involves sophisticated belief update reasoning logic where observations can have non-local implications requiring multiple reasoning steps (e.g., feedback that the “4” guessed at position 1 is not in the combination makes it possible to conclude that position 2 must be “6”, which then excludes “6” from the posterior belief at position 3), and QA involves extreme long-horizon generalization (from 2 to 16 objectives, and 6 to 20 steps) and much lengthier contexts (over 300 words in each piece of environment feedback).

**Novelty**

Some reviewers also questioned novelty compared to recent work on context summarization. We have added the mentioned works to our related work section, however, they:

- rely on querying an external memory store, which is an orthogonal approach that faces a very different set of constraints and can be combined with ABBEL for greater gains,

- or summarize static contexts in the single-step setting (e.g., MemAgent [2] divides a document into 5000-token chunks and iteratively summarizes them before taking one action), which doesn't apply to our multi-step settings where the initial context is short and grows long over many short steps, and have no method to update summaries on new actions and observations. Multi-step agents must also reason over their beliefs to select actions to gather missing information; whereas an agent in a single-step environment need not reason about this because all information is already given.

We highlight that MEM1[2], the only other work we are aware of addressing context summarization in the multi-step setting, is **also concurrently under review at ICLR** and does not perform as well as ABBEL in their own QA domain. Our approach of separating the belief from the reasoning is more principled and steerable than MEM1’s approach of using the full reasoning trace itself as the memory, allowing for our novel contributions of belief grading and further context compression through belief length penalties. Since our original submission, we removed normalization of the belief length penalty, thus allowing the magnitude of the length penalty to naturally decrease relative to outcome rewards as the beliefs get shorter, which led to ABBEL-Length-Penalty outperforming MEM1 by an even wider margin on both task performance and memory usage (please see our revised manuscript).

In addition to the new experiment, feedback from the reviewers on the writing has helped us improve the clarity of the paper in many ways. We detail how this has been done in individual responses to the reviewers.

[1] Zhou, Yifei, et al. "Sweet-rl: Training multi-turn llm agents on collaborative reasoning tasks."

[2] Yu, Hongli, et al. "MemAgent: Reshaping Long-Context LLM with Multi-Conv RL-based Memory Agent."

[3] Zhou, Zijian, et al. "MEM1: Learning to Synergize Memory and Reasoning for Efficient Long-Horizon Agents."

---

> ### Comment · Reviewer_dYtH · 2025-11-25
> **ColBench is not realistic**
>
> Thanks for the updates. I appreciate the new experiment, but I did want to mention here, that I think its quite a stretch to call ColBench a realistic benchmark.
>
> I'll leave the links here for other reviewers to evaluate for themselves: https://arxiv.org/abs/2503.15478 https://huggingface.co/datasets/facebook/collaborative_agent_bench/viewer/default/train?row=0 while providing my understanding of the dataset.
>
> The dataset consists of python functions generated by Llama 3.1 70B based on web documents. In most cases, these web documents have nothing to do with a programming problem, so one could consider the tasks entirely hallucinated by Llama 3.1 70B. Its unclear whether any of these tasks are useful, correct (test cases are also generated by Llama), or interesting, and they were never validated with a human study.
>
> That said, the experiment might still be valuable. I request the authors to not characterize it as "realistic", though you could say its more complex than the toy tasks presented earlier.

---

> > ### Author Response · Authors · 2025-11-26
> >
> > We thank reviewer dYtH for their feedback, and we see their point. We have updated how we refer to ColBench in our responses and in the manuscript. Though somewhat synthetic, we do believe ColBench is valuable as it is a computationally tractable testbed inspired by the very popular current use case of LLM assistants in interactive coding environments, where context summarization is an open problem [1].
> >
> > [1] OpenHands Context Condensensation for More Efficient AI Agents https://openhands.dev/blog/openhands-context-condensensation-for-more-efficient-ai-agents

---

### Author Response · Authors · 2025-12-04
**Final comment**

We summarize the reviews and ensuing developments for the AC.

In their initial reviews, reviewers stated that we tackle an “interesting” (y4kX, dYtH) and “important” (dYtH) problem, judge our solution to be “practical” (eHNU) and “simple yet effective” (icj6), and praise our “clear” presentation (y4kX, dYtH). We are grateful to reviewer dYtH for a productive discussion, which helped us clarify why most prior LLM context summarization work only applies to single-step settings and highlight how our work addresses a more general, and important yet understudied setting.

We have now addressed all major concerns raised by the reviewers with the following experiments, writing changes and clarifications:

**Concern:** is ABBEL also effective in more complex environments? (Icj6, eHNU, y4kX, dYtH)

Response: we show ABBEL is effective in a more complex environment, the collaborative coding benchmark ColBench: after RL, ABBEL achieves 88% the performance of the trained full-context model (Vanilla) while using only 49% of the memory (Section 5.2.3).

**Concern:** how does belief grading work in settings where the ground truth posterior is unavailable? (Icj6, eHNU, dYtH)

Response: we proposed a fully domain-general belief grading function that does not rely on ground truth posteriors (see overall response and Section 5.2.3 for details), and demonstrate it helps ABBEL learn significantly faster in ColBench, taking half as many steps to reach ABBEL's peak performance, and performing on-par with Vanilla while using only 44% as much memory at step 50/100.

**Concern:** missing baselines in QA: RL with the full context (eHNU), zero-shot performance of Qwen 7B models (y4kX)

Response: we trained a full-context RL baseline in QA (Vanilla), finding that it only performs slightly better than ABBEL, with no performance advantage at the 16 objective setting despite access to the full context and using 9.5x as much memory. We additionally evaluate zero-shot performance of ABBEL and Vanilla in QA, showing that RL training was highly effective in increasing performance. (Section 5.2.2)

**Concern:** not enough explanations and detail in the RL section (Icj6, dYtH)

Response: we condensed the frontier model eval section (section 4) and added more explanations to the RL section including a very thorough explanation of belief grading (section 5.1.2) and more detailed descriptions of the environments used (section 5.2).

**Concern:** missing LLM context summarization citations and baselines (eHNU, dYtH)

Response: we added the mentioned works to our related work section (Section 2), explaining why they can’t be applied to our multi-step interaction setting (see our discussion with reviewer dYtH for more), or are orthogonal approaches (reliant on external memory) that can be combined with ABBEL.

**Concern:** very similar to MEM1 (eHNU)

Response: we highlighted that MEM1 is also concurrently under review at ICLR, ABBEL performs better using less memory and the isolated belief state is more principled and steerable, allowing for our novel contributions of belief grading & further context compression through belief length penalties. We have concerns about MEM1’s design beyond its performance, and note their training requires 5x more datapoints and 2.5x more rollouts  (see our first response to eHNU for more details).

**Concern:** was ABBEL-Length Penalty significantly better than MEM1 in QA? (y4kX)

Response: we removed normalization on the length penalty (similar to [1], details in Section 5.1.1) and retrained ABBEL LP, finding it outperforms MEM1 by a much wider margin. (Exact Match Score 3.43 $\pm$0.05 for ABBEL LP, 2.50 $\pm$0.06 for MEM1) while still using much less memory (Peak Tokens 764$\pm$4 for ABBEL LP, 1058$\pm$7 for MEM1). See Section 5.2.2

Thanks to the reviewer feedback, we have significantly strengthened our empirical evidence for ABBEL’s effectiveness and generalizability, and improved the clarity of the paper.

[1] Arora, Daman, and Andrea Zanette. "Training language models to reason efficiently." arXiv preprint arXiv:2502.04463 (2025).

---

### Meta-Review · Area_Chair_ShRs · 2026-01-05

**Summary:**

The paper proposes ABBEL (Acting through Belief Bottlenecks Expressed in Language), replacing growing interaction histories with concise natural-language belief states for multi-step LLM tasks. Zero-shot frontier models lag full-context baselines due to belief-update errors. RL post-training, especially with belief grading, recovers performance and enables memory-accuracy trade-offs, outperforming full-context in synthetic tasks and nearing it in collaborative coding (ColBench) with less memory.

Reviewers praised the interpretable bottleneck, potential for efficient long-horizon agents, and RL innovations (belief grading, length penalties). However, concerns included limited novelty (similar to learned-memory works like MEM1), reliance on task-specific grading limiting generalization, synthetic/toy environments reducing real-world relevance, no human baselines initially, and unclear advantages over long-context RL.

The authors' rebuttal introduced a domain-general reconstruction-based grader, ColBench results (near full-context performance with half memory), human baselines (LLMs lag humans in strategic reasoning) and ablation analysis.

**Reviewer Concerns:**

Technical/methodological issues (grading generalization, baselines) were largely resolved. Remaining: perceived incrementality over concurrent memory works, depth of insight from synthetic settings, and broader validation needs.

**Reviewer Scores:**

Initial scores borderline (two at 4, one at 2). Rebuttal yielded engagement and one upward revision (to 6); consensus shifted toward weak reject.

---

### Decision · Program_Chairs · 2026-01-26

Reject